# CONGO:
## Compressive Online Gradient Optimization

**Jeremy Carleton**[1][*]   **Prathik Vijaykumar**[1]   **Divyanshu Saxena**[2]   **Dheeraj Narasimha**[3]
**Srinivas Shakkottai**[1]   **Aditya Akella**[2]
[1] Texas A&M University, [2] The University of Texas at Austin, [3] Inria

### ABSTRACT

We address the challenge of zeroth-order online convex optimization where the objective function's gradient exhibits sparsity, indicating that only a small number of dimensions possess non-zero gradients. Our aim is to leverage this sparsity to obtain useful estimates of the objective function's gradient even when the only information available is a limited number of function samples. Our motivation stems from the optimization of large-scale queueing networks that process time-sensitive jobs. Here, a job must be processed by potentially many queues in sequence to produce an output, and the service time at any queue is a function of the resources allocated to that queue. Since resources are costly, the end-to-end latency for jobs must be balanced with the overall cost of the resources used. While the number of queues is substantial, the latency function primarily reacts to resource changes in only a few, rendering the gradient sparse. We tackle this problem by introducing the Compressive Online Gradient Optimization framework which allows compressive sensing methods previously applied to stochastic optimization to achieve regret bounds with an optimal dependence on the time horizon without the full problem dimension appearing in the bound. For specific algorithms, we reduce the samples required per gradient estimate to scale with the gradient's sparsity factor rather than its full dimensionality. Numerical simulations and real-world microservices benchmarks demonstrate CONGO's superiority over gradient descent approaches that do not account for sparsity.

## 1 INTRODUCTION

Online convex optimization (OCO) plays an important role in managing distributed applications like web caching, advertisement placement, portfolio selection, and recommendation systems. These applications involve an unknown, time-varying convex system cost function $f_t(\boldsymbol{x}_t)$, dependent on a $d$-dimensional control $\boldsymbol{x}_t$, which the user selects in an online manner. Of particular interest are systems with the following key characteristics: (i) $f_t(\cdot)$ and its gradient are unknown, and obtaining measurements for any control $\boldsymbol{x}_t$ is costly; (ii) the gradient of $f_t(\cdot)$ is sparse, where only a few dimensions of $\boldsymbol{x}_t$ significantly affect $f_t(\cdot)$; and (iii) $f_t(\cdot)$ evolves slowly, allowing for multiple measurements between updates to $\boldsymbol{x}_t$. These features arise in many real-world settings, such as in queueing systems in communication, manufacturing, and distributed computing, where only a few bottlenecks exist at any given time and they must be identified through only a few measurements. Our goal is to develop efficient online gradient descent algorithms tailored to such scenarios.

A core challenge in OCO is that both the objective function and its gradient are usually unavailable in closed form. Thus, we must perform zeroth-order OCO, where gradients are estimated from objective function observations at specific points close to the current input. We will show that when gradients exhibit sparsity (see Definition 1), efficient gradient estimation is possible by combining simultaneous perturbation stochastic approximation (SPSA; Spall (1992)) and compressive sensing, achieving dimension-independent sub-linear regret scaling in OCO problems.

---

[*]Corresponding author: jcarleton@tamu.edu
Code available at https://github.com/5-Jeremy/CONGO

**Related Work on Compressive Sensing for Stochastic Optimization:** Introduced by Candes and Tao (2006), compressive sensing is widely used in signal processing for sparse signal recovery (Eldar and Kutyniok, 2012). Its value lies in requiring only logarithmic measurements relative to the vector's dimension $d$ to approximate an $s$-sparse vector; see Foucart and Rauhut (2013) for details.

To apply compressive sensing techniques, gradient measurements must be taken at randomly perturbed points, similar to SPSA. This idea was explored in (Borkar et al., 2018) in the context of stochastic optimization for a single unchanging objective function. Here, the gradient is measured as $\mathbf{A}\nabla f(\boldsymbol{x}) + \boldsymbol{e}$, with $\mathbf{A}$ being a Gaussian matrix and $\boldsymbol{e}$ an error term due to interference from other dimensions and high order terms in the Taylor approximation to the perturbed function value. Two function evaluations suffice per measurement, though the error requires averaging over multiple perturbations to ensure an acceptable level of accuracy of the gradient estimate.

Other significant work that leverages compressive sensing for stochastic optimization of a given objective function is (Cai et al., 2022), which introduces the Zeroth-Order Regularized Optimization (ZORO) algorithm. ZORO uses a Bernoulli measurement matrix and applies only one row of the matrix at a time. This method reduces the noise in gradient estimation, enabling accurate gradient estimates with fewer function evaluations.

More details of related work are presented in Appendix A.

**Our Contributions – CONGO:** We introduce the Compressive Online Gradient Optimization (CONGO) framework, applying compressive sensing methods to OCO for estimating gradients using a limited number of function evaluations at each online optimization step. Our goals for CONGO in the sparse OCO problem are: (i) develop a sample-efficient gradient estimation approach, (ii) analyze its regret over finite time, and (iii) empirically validate it in a general setting, and specifically for a real-world use case in a containerized microservice application.

We propose three CONGO variants: CONGO-B (leveraging ideas from Borkar et al. (2018)), CONGO-Z (leveraging ideas from the ZORO approach of Cai et al. (2022)), and a more effective sampling approach using a Gaussian measurement matrix, which we name CONGO-E. One of our key analytical results is to allow for a bound on the expected regret to be proven in spite of the chance for compressive sensing to fail to recover the gradient with low error. We prove that CONGO-Z and CONGO-E have a better sample complexity than CONGO-B for generating gradient estimates, and we derive a dimension-independent $O(\sqrt{T})$ regret bound over $T$ time steps for these algorithms. For CONGO-Z and CONGO-E, the $\mathcal{O}\left(s \log\left(\frac{d}{s}\right)\right)$ samples per gradient estimate greatly improves upon the $d + 1$ samples required for algorithms that do not consider sparse gradients (Agarwal et al., 2010).

We validate CONGO in three settings: (i) a fully stochastic setting where a new high-dimensional random objective function is sampled after each gradient descent step (Section 6); (ii) a a queueing system (under the Jackson network model), where resource allocations are chosen to balance processing time and resource cost (Section 7.1); and (iii) a real-world microservice autoscaler, where CPU resources are allocated to containers of a benchmark application (Gan et al., 2019) (Section 7.2). In setting (i), CONGO-E and CONGO-Z perform similar to gradient descent with full gradient information, thus matching our expectations from the theoretical results. In settings (ii) and (iii), CONGO-E outperforms standard SPSA in most cases even in low-dimensional settings, with its performance improving as sparsity increases. In the microservice setting, CONGO-E approaches the performance of an optimal reinforcement learning (RL) baseline even under variable loads. We conclude that CONGO enables algorithms which effectively exploit sparsity.

## 2 PROBLEM FORMULATION

*System model and sequence of events:* We consider an online learning problem over $T$ rounds indexed from 1 to $T$. At the start of each round $t$, the controller must choose a $d$-dimensional control $\boldsymbol{x}_t \in \mathcal{K}$ where $\mathcal{K} \subset \mathbb{R}^d$ is known. We are specifically interested in the high dimensional regime such that $d$ is a large value. Next, an adversary chooses a differentiable convex function $f_t : \mathbb{R}^d \to \mathbb{R}$ and the controller incurs the cost $f_t(\boldsymbol{x}_t)$. The controller is *unaware of the function $f_t$*, however, we may acquire samples of $f_t$ in a local vicinity of $f_t(\boldsymbol{x}_t)$. It is assumed that these samples are costly to obtain, or that there is a limit to how many can be taken, such that we desire an algorithm to be as sample-efficient as possible. Note that in our constrained optimization setting, we permit the samples

used for gathering information on $\tilde{\nabla} f_t(\boldsymbol{x}_t)$ to lie outside of $\mathcal{K}$ so long as they remain close to $\boldsymbol{x}_t$ in expectation.

*Control strategy:* In this work, we will examine an *online projected gradient descent algorithm* where the samples retrieved are used to estimate a noisy gradient. In particular, at the end of round $t$, we use the samples taken to compute an estimate $\tilde{\nabla} f_t(\boldsymbol{x}_t)$. The online gradient descent algorithm dictates that we should use

$$\boldsymbol{x}_{t+1} = \underset{\boldsymbol{x} \in \mathcal{K}}{\operatorname{argmin}} ||\boldsymbol{x} - (\boldsymbol{x}_t - \eta_t \tilde{\nabla} f_t(\boldsymbol{x}_t))||_2 \tag{1}$$

as the control for our next time step. Here $\eta_t$ is a predetermined step-size chosen by the controller.

*Assumptions:* Our setting uses two key assumptions on the constraint set $\mathcal{K}$ and the observed functions.

**Assumption 1** (**Constraint Set Properties**). *The set $\mathcal{K} \subset \mathbb{R}^d$ satisfies the properties: (i) $\mathcal{K}$ is compact and convex; and (ii) $\sup_{\boldsymbol{x} \in \mathcal{K}} ||\boldsymbol{x}||_2 \leq R$.*

An important way in which our setting differs from standard online convex optimization is that we make an assumption about the sparsity of $\nabla f_t$. We define sparsity as follows:

**Definition 1** (**Sparsity**). *The support of a vector $\boldsymbol{x} \in \mathbb{R}^d$ is:*

$$supp(\boldsymbol{x}) := \{j \in \{1, 2, \ldots d\} : \boldsymbol{x}_j \neq 0\}. \tag{2}$$

*We call a vector $\boldsymbol{x}$ $s$-sparse if $|supp(\boldsymbol{x})| \leq s$.*

Our overall constraints on the set of functions the adversary can choose from are summarized below:

**Assumption 2** (**Function Properties**). *Let $\mathcal{F}$ denote the set of all functions $f$ which satisfy the properties: (i) $f$ is convex, $L_f$-Lipschitz, and $L$-smooth; and (ii) $\nabla f(\boldsymbol{x})$ is $s$-sparse $\forall \boldsymbol{x} \in \mathcal{K}$. For each time $t \in [1, T]$, we assume our functions $f_t$ are chosen such that $f_t \in \mathcal{F}$.*

As mentioned in Section 1, the significance of the sparsity assumption is that an algorithm may be able to exploit this sparsity such that the number of samples it requires and the overall regret has little or no dependence on $d$. The challenge, of course, is that we do not know *a priori* which dimensions of $\nabla f_t$ are nonzero at our current control point, and this can change at every round.

*Performance Metric:* Our performance metric of interest is the *adversarial regret* at the end of $T$ rounds, which measures the difference in the cost accumulated over the $T$ rounds with that of the best fixed control point $\boldsymbol{x}^*$ in hindsight. In what follows we shall formally define the adversarial regret.

**Definition 2** (**Adversarial Regret**). *Given a sequence of convex functions $\{f_t\}_{t=1}^T$ satisfying Assumption 2, our objective is to minimize the following quantity which we call the regret:*

$$R(T) := \max_{f_\tau \in \mathcal{F}, \tau=1,2,\ldots T} \left( \sum_{t=1}^T f_t(\boldsymbol{x}_t) - \min_{\boldsymbol{x}^*} \sum_{t=1}^T f_t(\boldsymbol{x}^*) \right) \tag{3}$$

## 3 MOTIVATING EXAMPLE

The Jackson network control problem is one possible application of CONGO and relates to the real-world problem of autoscaling microservices. In a Jackson network (Srikant and Ying, 2014), jobs enter a set of interconnected queues. When de-queued, jobs are processed by a server and then may re-enter the queue, jump to another, or exit the system. The (end-to-end) latency is the delay between job arrival and completion and is affected by the service rates of the servers. Our goal is to minimize the average latency while balancing resource costs, which grow linearly with increased resource allocation. The average service time in a Jackson network is a convex, non-increasing function of allocated resources; meanwhile, the workload changes over time, making this an OCO problem.

Let $L(\boldsymbol{\lambda}_t, \boldsymbol{x}_t)$ denote the expected end-to-end latency for a given workload $\boldsymbol{\lambda}_t$ and allocation $\boldsymbol{x}_t$. The objective function (total cost) that we wish to minimize is:

$$f_t(\boldsymbol{x}_t) = L(\boldsymbol{\lambda}_t, \boldsymbol{x}_t) + w \sum_{i=1}^d [\boldsymbol{x}_t]_i, \tag{4}$$

where $w$ is the price per unit of allocated resources (assumed uniform across queues). Note that each element of the gradient of $L(\boldsymbol{\lambda}_t, \cdot)$ corresponds to a different queue. We can only sample latencies

for a finite number of jobs to limit measurement overhead, and these samples are used to estimate the gradient. The service cost is known, so only the gradient of $L(\boldsymbol{\lambda}_t, \cdot)$ needs to be estimated. If the system is bottlenecked only at a few queues, the problem will be sparse, with only a few large gradients, which we must identify using a limited number of measurements.

Such sparsity is observed in large-scale heterogeneous microservices applications. Similar to the Jackson network model, requests in microservice applications queue at multiple microservices before completion (Gan et al., 2019). Resource allocation to microservices, e.g. CPU resources, must be carefully managed to achieve tight latency targets. Appendix A includes a review of microservice literature and the challenges of resource allocation, particularly the curse of dimensionality arising from numerous services and flows (production microservice graphs may have up to 100s of microservices (Huye et al., 2023)). Typically, only a few microservices are bottlenecks, and adjusting resources for any microservice which is not a bottleneck will not significantly impact $L(\boldsymbol{\lambda}_t, \boldsymbol{x}_t)$ (Qiu et al., 2020), leading to an approximately sparse gradient as per Definition 1, which we have confirmed experimentally (see Fig. 1). This sparse bottleneck phenomenon has been observed in other offline-trained autoscalers (Qiu et al., 2020; Sachidananda and Sivaraman, 2024; Wang et al., 2024) that identify specific microservice instances that are bottlenecks and adjust their allocations. We also observe that $L$ is convex in both its arguments (Fig. 1). Thus, a scheme like CONGO can be expected to perform well at online resource management for microservice applications.

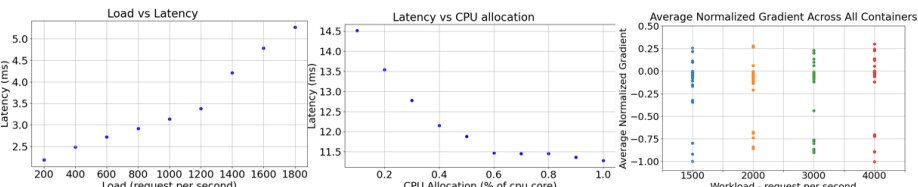

Figure 1: Left: Convexity of latency vs. CPU allocated to the nginx-thrift container in the SocialNetwork benchmark. Center: Convexity of latency vs. arrival rate for compose-post requests. Right: Sparsity of average relative gradients of end-to-end latency for 26 containers under different offered workloads. Only a few show large (negative) gradients for each workload, making them good targets for increased CPU allocation.

## 4 CONGO FRAMEWORK AND ALGORITHMS

The CONGO framework characterizes a class of online convex optimization algorithms that consist of four steps: (i) a measurement matrix generation step, (ii) a step where measurements are collected, (iii) a constrained gradient recovery step, and (iv) a gradient descent update step. CONGO algorithms can be broken into an inner loop that estimates the gradient (steps (i)-(iii)) and an outer loop that performs gradient descent (step (iv)). Since no existing OCO approach utilizes compressive sensing, we make decisions for each of the above steps, leveraging insights from work on stochastic optimization with a fixed objective. As we do so, we present variants CONGO-B and CONGO-Z, which are motivated by Borkar et al. (2018) and Cai et al. (2022), along with a more efficient variant, CONGO-E. Some of the design choices we made for the OCO setting are explained in greater detail in Appendix D.

**(i) Measurement Matrix Generation Method:** To obtain the gradient approximation $\tilde{\nabla} f(\boldsymbol{x})$, we first obtain "compressed" measurements of the form $\boldsymbol{y} = \mathbf{A} \nabla f(\boldsymbol{x}) + \boldsymbol{e}$ (where the range space of $\mathbf{A}$ is of lower dimension than the domain space). We apply the theory of compressive sensing to guide our choice for the matrix $\mathbf{A}$ used in the simultaneous perturbation step (step (ii)). The recovery guarantees we use require $\mathbf{A}$ to satisfy the $2s^{\text{th}}$ or $4s^{\text{th}}$ restricted isometry property (RIP) as defined in Appendix A. This property approximately preserves the notion of distance in the domain and range of the transformation represented by the matrix $\mathbf{A}$ for sufficiently sparse vectors. Since we will eventually create $\tilde{\nabla} f(\boldsymbol{x})$ from the measurements (step iii), we need $\mathbf{A}$ to be such that the information in $\nabla f(\boldsymbol{x})$ can be recovered from $\mathbf{A} \nabla f(\boldsymbol{x})$.

It is known (Foucart and Rauhut, 2013) that matrices drawn from dense random distributions like Gaussian matrices (with $\mathcal{N}(0, 1)$ entries) and Bernoulli matrices (with Rademacher entries) give the best guarantees for satisfying the RIP, so those matrices are used in the CONGO-style algorithms we consider. Since these matrices may fail to satisfy the RIP (with a small probability), CONGO-style algorithms resample the measurement matrix at each iteration to ensure that compressive sensing failures are limited to single iterations.

**(ii) Measurement Procedure:** The measurements of the form $\mathbf{A}\nabla f(\boldsymbol{x}) + \boldsymbol{e}$ are obtained using variants of the simultaneous perturbation stochastic approximation (SPSA). The original idea for SPSA is credited to Spall (1992). We introduce it here with more details in Appendix A. First, generate a sequence of $d$ i.i.d. Rademacher random variables, $\{\Delta_j\}_{j=1}^d$. Next, generate, a perturbation vector $\boldsymbol{\Delta} := \sum_{j=1}^d \boldsymbol{u}_i \Delta_j$ where $\boldsymbol{u}_i$ is the unit vector along the $i$th dimension. Then we compute gradient estimates along each dimension:

$$\frac{f(\boldsymbol{x} + \delta\boldsymbol{\Delta}) - f(\boldsymbol{x})}{\delta\Delta_j} \approx \frac{\partial f(\boldsymbol{x})}{\partial x_i} + \sum_{i \neq j} \frac{\partial f(\boldsymbol{x})}{\partial x_i} \frac{\Delta_i}{\Delta_j}. \tag{5}$$

These $d$ measurements combine to give an estimate of $\nabla f(\boldsymbol{x})$. When applying compressive sensing, we seek measurements which form a vector approximately equal to $\mathbf{A}\nabla f(\boldsymbol{x})$ where $\mathbf{A} \in \mathbb{R}^{m \times d}$ with $m << d$. This requires us to form perturbations using the rows of $\mathbf{A}$, either by replacing $\boldsymbol{\Delta}$ with a row $\mathbf{a}_i$ and repeating $m$ times or by replacing $\boldsymbol{\Delta}$ with $\mathbf{A}^T \Delta$ to form a linear combination. While the latter method only requires two function evaluations, it introduces a noise term due to interference between measurements for different rows; this was observed previously by Borkar et al. (2018).

To see how these methods give us an approximation of $\mathbf{A}\nabla f(\boldsymbol{x})$, note that since $f$ is differentiable, the Taylor approximation for $f(\boldsymbol{x} + \delta\boldsymbol{\Delta})$ with $\boldsymbol{x}$ as the reference point is as shown in equation 6 after cancelling out terms:

$$f(\boldsymbol{x} + \delta\boldsymbol{\Delta}) = f(\boldsymbol{x}) + \langle \nabla f(\boldsymbol{x}), \delta\boldsymbol{\Delta} \rangle + \xi \tag{6}$$

where $\xi$ represents the Lagrange remainder of the first-order Taylor approximation. In particular, we have the bound

$$\xi \leq \frac{1}{2} ||\nabla^2 f||_2 ||\delta\boldsymbol{\Delta}||_2^2 \tag{7}$$

Thus, $(f(\boldsymbol{x} + \delta\boldsymbol{\Delta}) - f(\boldsymbol{x}))/\delta = \langle \nabla f(\boldsymbol{x}), \boldsymbol{\Delta} \rangle + \mathcal{O}(\delta)$.

**Composition of $\boldsymbol{y}$ from measurements:** If $\boldsymbol{\Delta} = \mathbf{a}_i$, then the first term is exactly want we want for $y_i$. However, if $\boldsymbol{\Delta} = \mathbf{A}^T \Delta$, then we can divide by $\Delta_i$ to isolate $\langle \nabla f(\boldsymbol{x}), \mathbf{a}_i \rangle$, but there will be other terms of the form $\Delta_i^{-1} \langle \nabla f(\boldsymbol{x}), \Delta_j \mathbf{a}_j \rangle$ left (if the elements of $\Delta$ are Rademacher random variables, then these terms will have zero mean). Since any difference between $y_i$ and $\langle \nabla f(\boldsymbol{x}), \mathbf{a}_i \rangle$ adds to the error in the final estimate of $\nabla f(\boldsymbol{x})$, algorithms using the second method may need to repeat the process several times with different random draws of $\Delta$ and average over the measurements to control this interference between measurements. In either case, the CONGO framework also requires replacing $\delta$ by $\delta ||\boldsymbol{\Delta}||_2^{-2}$ to cancel out the factor of $\boldsymbol{\Delta}$ in the Lagrange remainder. This is important because if a factor of $||\boldsymbol{\Delta}||_2^2$ is left in the error term, its expectation will be present in the regret bound, and hence the regret bound will no longer be dimension-independent.

The difference between the two measurement schemes is made clear in the following lemma, which bounds the overall measurement error (prior to any potential averaging):

**Lemma 1.** *Given a matrix $\mathbf{A} \in \mathbb{R}^{m \times d}$, $\delta > 0$, and a function $f$ which is convex, $L_f$-Lipschitz, and $L$-smooth, if a vector of measurements $\boldsymbol{y}$ is constructed as*

$$y_i = \left( f(\boldsymbol{x} + \frac{\delta}{||\mathbf{a}_i||_2^2} \mathbf{a}_i) - f(\boldsymbol{x}) \right) \frac{||\mathbf{a}_i||_2^2}{\delta} \quad \forall i \in [m],$$

*then*

$$\boldsymbol{y} = \mathbf{A}\nabla f(\boldsymbol{x}) + \boldsymbol{e}$$

*where $||\boldsymbol{e}||_2 \leq \frac{1}{2}\delta ||\nabla^2 f||_2 \sqrt{m} \leq \frac{L}{2}\delta\sqrt{m}$.*
*If instead, $\boldsymbol{y}$ is constructed as*

$$y_i = \left( f(\boldsymbol{x} + \frac{\delta}{||\mathbf{A}^T\Delta||_2^2} \mathbf{A}^T\Delta) - f(\boldsymbol{x}) \right) \frac{||\mathbf{A}^T\Delta||_2^2}{\delta\Delta_i} \quad \forall i \in [m],$$

*where $\Delta \in \mathbb{R}^m$ is a vector of Rademacher random variables independent from the entries of $\mathbf{A}$, then*

$$\boldsymbol{y} = \mathbf{A}\nabla f(\boldsymbol{x}) + \boldsymbol{e}_1 + \boldsymbol{e}_2$$

*with $||\boldsymbol{e}_2||_2 \leq \frac{1}{2}\delta ||\nabla^2 f||_2 \sqrt{m} \leq \frac{L}{2}\delta\sqrt{m}$ and*

$$[\boldsymbol{e}_1]_i = \sum_{j \neq i} \frac{\Delta_j^l \langle \nabla f(\boldsymbol{x}), \mathbf{a}_j \rangle}{\Delta_i^l}$$

**(iii) Constrained Gradient Recovery:** Many recovery methods are available that estimate a sparse vector $\boldsymbol{v}$ given measurements of the form $\mathbf{A}\boldsymbol{v}+\boldsymbol{e}$ under suitable conditions on $\mathbf{A}$. Two of these already have well-studied guarantees: (i) Basis Pursuit (Chen et al., 1998) and (ii) Compressive Sampling Matching Pursuit (CoSaMP) (Needell and Tropp, 2008). See Appendix A for an introduction. Note that the recovery guarantees we use in Section 5 require us to scale $\mathbf{A}$ and $\boldsymbol{y}$ by $\frac{1}{\sqrt{m}}$ where $m$ is the number of rows of $\mathbf{A}$ before performing the recovery. Regardless of which method is used, the CONGO framework requires that $||\tilde{\nabla} f_t(\boldsymbol{x}_t)||_2^2$ be explicitly bounded over all rounds, and to ensure this we either introduce an explicit constraint (in the case of basis pursuit) or perform post-processing. If the recovery procedure fails to return a vector satisfying the expected bound, we replace it with the zero vector. Although the choice of the zero vector is not critical to the regret bound, it makes some practical sense since it avoids the risk of unexpectedly moving to a much worse control point.

**(iv) Gradient Descent Update Step** In this paper, we consider constrained OCO (see Assumption 1). Thus, our gradient descent update takes the form of equation 1.

## 4.1 Construction of CONGO Variants

We construct three variants of CONGO based on some of the choices that one can make for each step.

**CONGO-B** uses a Gaussian measurement matrix and a combination of the rows of the measurement matrix for each perturbation. Averaging is applied to mitigate the effect of interference between measurements. Basis pursuit is applied to obtain the gradient estimate. The pseudocode for this algorithm is presented in Appendix B.

**CONGO-Z** uses a Bernoulli measurement matrix and single-row perturbations resulting in exactly $m + 1$ function evaluations per gradient estimate (since no averaging is applied). CoSaMP is used to obtain the gradient estimate. Due to the similarity between CONGO-Z and CONGO-E, we do not include the pseudocode for CONGO-Z here.

**CONGO-E** modifies CONGO-Z by using a Gaussian measurement matrix, which we find leads to performance improvements across all of our experiments. We chose to stick with single-row perturbations to avoid the interference between measurements, since our analytical results show that this is necessary in order to obtain the desired regret bound with an efficient number of samples. We maintain CoSaMP as the recovery method for practical reasons; see Appendix E. The pseudocode for CONGO-E is provided in Algorithm 1.

**Remark 1.** *As alluded to above, the gradient estimation procedure used for CONGO-B introduces additional error due to interference since the measurements for every row of $\mathbf{A}$ are being compressed into a single sample. We have found that this leads to weaker theoretical results and worse empirical performance (even with averaging) than CONGO-Z/E. Our intention with CONGO-B is to illustrate the importance of certain design choices by comparing it to CONGO-Z/E.*

## 5 Theoretical results

In this section, we first present the general theorem used to prove CONGO regret bounds. Then, we state the bounds achieved by CONGO-B, CONGO-Z and CONGO-E. We establish that all three algorithms achieve the optimal regret scaling of $\mathcal{O}(\sqrt{T})$ without a dependence on the dimension. For CONGO-Z and CONGO-E in particular, this only requires $\mathcal{O}(s \left(\log \left(\frac{d}{s}\right) + \log(2T)\right))$ samples per round. In the non-sparse case it has been shown (Agarwal et al., 2010) that this can be achieved using $d + 1$ samples per round, but by taking advantage of the sparsity we are able to do so using a number of samples that depends only logarithmically on $d$. All proofs are postponed to Appendix C.

The following is the key theorem for the CONGO framework:

**Theorem 1.** *Let assumptions 1 and 2 hold, and further assume that $\forall t \in [T]$:*

- $||\tilde{\nabla} f_t(\boldsymbol{x}_t) - \nabla f_t(\boldsymbol{x}_t)||_2 \leq M\delta$ *with probability at least* $1 - \frac{1}{T}$

- $||\tilde{\nabla} f_t(\boldsymbol{x}_t)||_2 \leq M\delta + L_f$

---

**Algorithm 1** CONGO-E: Compressive Online Gradient Optimization - Efficient Version

---

**Require:** Constraint set $\mathcal{K}$, Horizon $T$, sparsity $s$, Lipschitz constant $L_f$, Smoothness parameter $L$, Parameters $(\eta_t)_{t=1}^T$, $\delta$, $m$, Initialization point $\boldsymbol{x}_1$

  **for** $t = 1, 2, \cdots, T$ **do**

    $\mathbf{A}^t \leftarrow$ randomly drawn $m \times d$ matrix with $\mathcal{N}(0,1)$ entries    $\triangleright$ *Generate measurement matrix*

    Query $f_t(\boldsymbol{x}_t)$

    **for** $i = 1, 2, \cdots, m$ **do**

      $\boldsymbol{\Delta} \leftarrow \frac{\delta}{\|\mathbf{a}_i^t\|_2} \mathbf{a}_i^t$

      Query $f_t(\boldsymbol{x}_t + \boldsymbol{\Delta})$

      $y_i \leftarrow (f_t(\boldsymbol{x}_t + \boldsymbol{\Delta}) - f_t(\boldsymbol{x}_t)) \frac{\|\mathbf{a}_i^t\|_2^2}{\delta}$      $\triangleright$ *Construct $\boldsymbol{y}$ from measurements*

    **end for**

    $\tilde{\nabla} f_t(\boldsymbol{x}_t) \leftarrow \text{CoSaMP}(\frac{1}{\sqrt{m}}\mathbf{A}^t, \frac{1}{\sqrt{m}}\boldsymbol{y}, s)$      $\triangleright$ *Constrained gradient recovery*

    **if** $\|\tilde{\nabla} f_t(\boldsymbol{x}_t)\|_2 > L_f + \frac{7.21}{2} L\delta$ **then**

      $\tilde{\nabla} f_t(\boldsymbol{x}_t) \leftarrow 0$

    **end if**

    $\boldsymbol{x}_{t+1} \leftarrow \underset{\boldsymbol{x} \in \mathcal{K}}{\text{argmin}} \|\boldsymbol{x} - (\boldsymbol{x}_t - \eta_t \tilde{\nabla} f_t(\boldsymbol{x}_t))\|_2$      $\triangleright$ *Gradient descent update*

  **end for**

  Incur regret $\sum_{t=1}^T f_t(\boldsymbol{x}_t) - f_t(\boldsymbol{x}^*)$

---

*where $M \geq 0$ is independent of $T$. Then with $\eta_t = \frac{1}{L_f\sqrt{T}} \forall t$ and $\delta = \frac{1}{MT}$, the regret of an algorithm running projected gradient descent using the estimated gradient $\tilde{\nabla} f_t(\boldsymbol{x}_t)$ after $T$ rounds scales as $\mathcal{O}((D + \frac{R}{2} + \frac{4R}{L_f})L_f\sqrt{T})$ where $R$ is as in Assumption 1 and $D = \sup_{\boldsymbol{x},\boldsymbol{y} \in \mathcal{K}} \frac{1}{2}\|\boldsymbol{x} - \boldsymbol{y}\|_2^2$ (see proof for the full expression).*

The following lemma uses known guarantees on the restricted isometry constants (RICs) for Gaussian and Bernoulli random matrices to establish recovery guarantees. In turn, this will be used to show conditions under which our CONGO-style algorithms satisfy the assumptions of Theorem 1.

**Lemma 2.** *On any round $t$, Given a Gaussian or Bernoulli random matrix $\mathbf{A}^t$ and measurements $\boldsymbol{y} = \mathbf{A}^t\nabla f_t(\boldsymbol{x}_t) + \boldsymbol{e}$, if $\mathbf{A}^t$ and $\boldsymbol{y}$ are first rescaled by $\frac{1}{m}$, then under Assumption 1, the gradient estimate $\tilde{\nabla} f_t(\boldsymbol{x}_t)$ returned by CoSaMP or basis pursuit satisfies*

$$\|\tilde{\nabla} f_t(\boldsymbol{x}_t) - \nabla f_t(\boldsymbol{x}_t)\|_2 \leq C\|\boldsymbol{e}\|_2$$

*with probability at least $1 - \epsilon$ so long as*

$$m \geq C'\left(\kappa s \log\left(\frac{ed}{\kappa s}\right) + \log(2\epsilon^{-1})\right),$$

*where C, C', and $\kappa$ are constants depending on the type of measurement matrix used and the recovery algorithm (but not on $d$ or $s$). In particular, one can use $C = 3$ and $\kappa = 2$ for basis pursuit and $C = 7.21$ and $\kappa = 4$ for CoSaMP; see Remark 2 for a note on the value of $C'$.*

Based on Lemma 2 and Theorem 1, $m$ must depend logarithmically on $T$ to ensure the guarantee. While this dependence is weak, adjusting the number of samples as the problem horizon increases is undesirable. Fortunately, in all our experiments, we could omit the $T$-dependent term in $m$ and still achieve strong performance.

We now give the theoretical result for CONGO-B (Algorithm 2), and then show how it is improved upon by CONGO-Z and CONGO-E in terms of the number of function evaluations per round.

**Theorem 2.** *Let assumptions 1 and 2 hold. Then CONGO-B with $\eta = \frac{1}{L_f\sqrt{T}}$, $\delta = \frac{1}{3LT}$, $m = \left\lceil C\left(2s\log\left(\frac{ed}{2s}\right) + \log(4T)\right)\right\rceil$ (for a constant $C$ independent of $s$, $d$, and $T$) has an expected regret, which, after $T$ rounds, scales as $\mathcal{O}((D + \frac{R}{2} + \frac{4R}{L_f})L_f\sqrt{T})$. Furthermore, the average number of samples which the algorithm must take on a round to achieve this is no greater than $36d(m-1)^2 L_f^2 \log(2m)T\log(2mT) + 2$.*

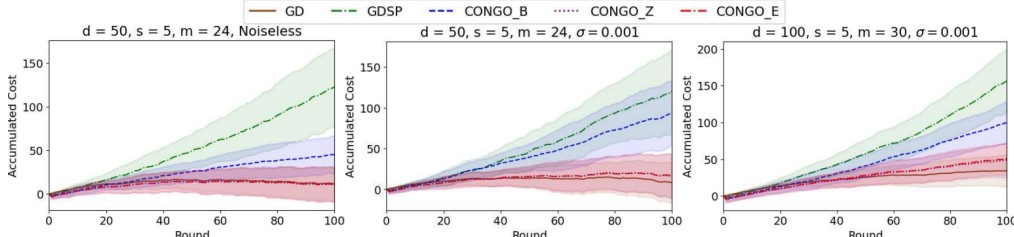

Figure 2: Comparison of accumulated costs, where GD represents gradient descent with full gradient information. Left: Noiseless function evaluations with $d = 50$. Center: Noisy evaluations with $d = 50$. Right: Noisy evaluations with $d = 100$. $\sigma$ represents the noise standard deviation. Even with noise, CONGO-Z and CONGO-E perform closely to the GD baseline.

Clearly, the number of samples which CONGO-B demands for this guarantee is not realistic for practical scenarios. However, it should be noted that due to the loose upper bounds used to obtain this value, the situation appears much worse than it actually is. Indeed, in the numerical experiments of Section 6 we find that when the number of samples taken is only $cm$ for a small constant $c$, the performance from CONGO-B is still significantly better than the SPSA algorithm that does not account for sparsity. Nonetheless, we would like to have an algorithm which can give the desired regret bound with a reasonable number of samples which is significantly less than $d$. The following result shows that CONGO-Z and CONGO-E, which only require $m + 1$ samples, suit this purpose.

**Theorem 3.** *Let assumptions 1 and 2 hold. Then CONGO-Z and CONGO-E with* $\eta = \frac{1}{L_f \sqrt{T}}$, $\delta = \frac{2}{7.2LT}$, $m = \left\lceil C \left( 4s \log \left( \frac{ed}{4s} \right) + \log(2T) \right) \right\rceil$ *(for a constant C independent of s, d, and T) have an expected regret which after T rounds scales as* $\mathcal{O}((D + \frac{R}{2} + \frac{4R}{L_f})L_f \sqrt{T})$.

## 6 BENEFITS OF COMPRESSIVE SENSING

In what follows, we empirically demonstrate the superiority of the compressive sensing approach to standard SPSA (GDSP denotes the algorithm which performs gradient descent using standard SPSA) in the online setting. We also include projected gradient descent with full gradient information (denoted GD) as an optimal baseline. In the numerical results presented in this section, we restrict the class of functions which the adversary can choose from to quadratic functions of the form

$$f(\boldsymbol{x}) = \boldsymbol{x}^T \boldsymbol{D} \boldsymbol{x} + \boldsymbol{b}^T \boldsymbol{x} + c \tag{8}$$

where $\boldsymbol{D}$ is a diagonal matrix. Importantly, the vectors $\text{diag}(\boldsymbol{D})$ and $\boldsymbol{b}$ are both $s$-sparse with identical support. This ensures that the gradient $\nabla f(\boldsymbol{x}) = 2\boldsymbol{D}\boldsymbol{x} + \boldsymbol{b}$ is $s$-sparse. The nonzero entries of $\boldsymbol{b}$ are sampled from a Gaussian distribution with a mean of -1 and a standard deviation of 1, and the nonzero entries of $\boldsymbol{D}$ are sampled from the same distribution with the absolute value taken to ensure that $\boldsymbol{D}$ is positive semidefinite. The value of $c$ is sampled from a folded normal distribution so that $c \geq 0$. The constraint set is a Euclidean ball centered on the origin with radius $R$. In the experiments for Fig. 2, we set $R = 100$.

Fig. 2 shows the results for three experiments with a moderately high level of sparsity. These plots are averaged over 50 random seeds, and the standard deviations are indicated with the shaded areas. CONGO-Z, CONGO-E, and GDSP all use $m + 1$ samples per gradient estimate, but CONGO-B uses $3m$ and $6m$ for $d = 50$ and $d = 100$, respectively. We first considered the case where function evaluations are noiseless, and set $\delta = 1\mathrm{e}^{-5}$. As expected, the CONGO-Z and CONGO-E algorithms achieved almost the same performance as exact gradient descent, while CONGO-B did worse since the amount of averaging was not enough to counteract the interference between dimensions. All of the CONGO-style algorithms outperformed GDSP. In the second and third experiments, we considered the case where noise from a $\mathcal{N}(0, 0.001)$ distribution was added to the function evaluations and found that CONGO-Z and CONGO-E still did not perform much worse than exact gradient descent. For CONGO-B, we had to increase the number of samples it averaged over for it to maintain a similar performance. We performed several additional experiments in this setting to investigate robustness; the results can be found in Appendix F. More details on the parameters used in the numerical simulations are given in Appendix G.

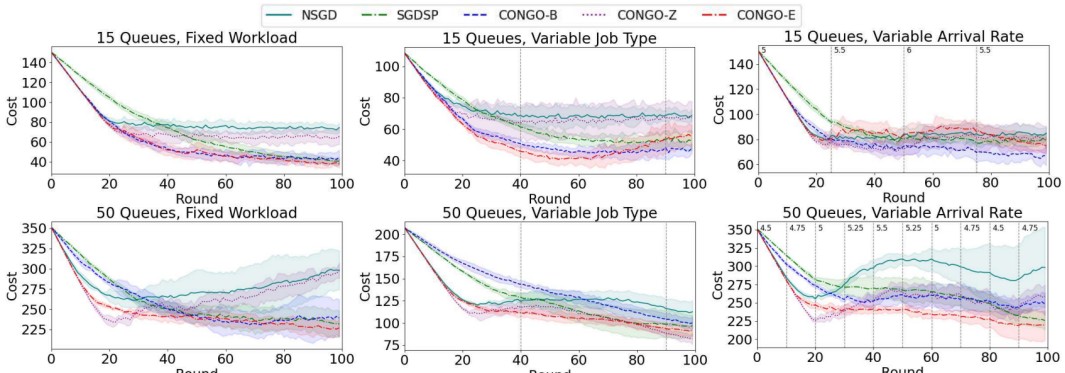

Figure 3: Jackson network simulations while varying the number of queues, workload, job types and arrival rates. The CONGO algorithms, especially CONGO-E, reduce cost faster than the baselines.

# 7 AUTOSCALING OF CONTAINERIZED MICROSERVICES

We now discuss the application of CONGO to the use-cases introduced in Section 3. In Section 7.1, we simulate Jackson networks under job distributions with unique routes, showing that the benefits of compressive sensing can be reaped in many scenarios. Then, in Section 7.2, we deploy our CONGO algorithms on a real-world microservice benchmark and demonstrate their practical benefits.

## 7.1 JACKSON NETWORK SIMULATION

**Methodology:** In the following experiments, we simulate the Jackson Network model introduced in Section 3. In the simulation, the service time experienced by jobs at a queue is inversely related to the allocation for that queue: given an allocation $x$ the service time at the $i$th queue is $\frac{1}{x_i+0.1}$. For simplicity, we set $w = 1$ in equation 4. We consider two different Jackson network layouts – one with 15 and one with 50 microservices. Details can be found in Appendix G. We conduct experiments for the two Jackson network layouts with three types of workloads: (i) fixed workload, (ii) fixed arrival rate with variable job distribution, and (iii) variable arrival rate with fixed job distribution.

**Baselines:** We compare the three CONGO-style algorithms with each other and two gradient-descent baselines to concretize the benefits of compressive sensing observed for this application. The first, Naive Stochastic Gradient Descent (NSGD), takes the naive approach of estimating the partial gradient along each dimension in isolation (thus requiring $d + 1$ samples per round). The second is the GDSP algorithm mentioned in Section 6 (now denoted SGDSP since it is applied to stochastic gradient descent). See Appendix G for the implementation details of the algorithms, including hyperparameters. The results for each of these algorithms are averaged over five random seeds.

**Results:** The results are shown in Fig. 3. First, we note that NSGD, despite using more samples on each round than the other algorithms, shows poor performance in every scenario; this is evidence for the effectiveness of simultaneous perturbation. We also note that while SGDSP eventually reaches a cost on par with that of CONGO-E in each scenario, it consistently takes much longer to reach that cost, leading to a significantly higher regret. In general, CONGO-E's advantage from quickly lowering its cost in the early rounds is amplified when there are more queues. We see that CONGO-B performs well in the limited-sparsity regime with only 15 queues, while in the high-dimensional regime it fails to outperform SGDSP (likely owing to greater interference). CONGO-Z performs poorly in many scenarios, which we believe is due to the Bernoulli random matrix being less effective for sparse recovery than the Gaussian random matrix for the same number of measurements.

## 7.2 REAL-WORLD MICROSERVICE BENCHMARK DEPLOYMENT

**Methodology:** We test the ability of CONGO-style algorithms to converge to an efficient resource allocation when autoscaling on a real-world benchmark. We utilize the SocialNetwork application from the DeathStarBench suite (Gan et al., 2019) which represents a small-scale social media platform with various request types such as 'compose-post', 'read-user-timeline', and 'read-home-timeline'.

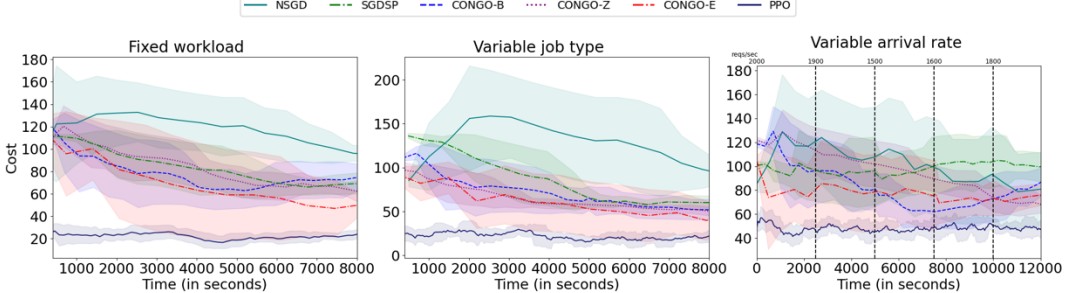

Figure 4: Overall costs in a microservice benchmark application under different workload patterns. CONGO-E consistently outperforms the baselines and overall performs better than CONGO-B and CONGO-Z, and it approaches the performance of a policy pre-trained for each scenario via PPO.

As with the experiments in Section 7.1, three scenarios are considered: (i) fixed workload, (ii) fixed arrival rate with variable job types, and (iii) variable arrival rate with fixed job types. As before, we use a combination of the end-to-end latency and a cost associated with CPU allocations as the overall cost function. If $\bar{L}(\cdot, \cdot)$ denotes the sample mean of the end-to-end latency, the cost is calculated as:

$$f_t(\boldsymbol{x}_t) = \psi \bar{L}(\boldsymbol{\lambda}_t, \boldsymbol{x}_t) + w \sum_{i=1}^{d} [x_t]_i$$

Note that for the purposes of minimizing the cost function, the introduction of the latency weight $\psi$ has the same effect on the optimal solution as rescaling $w$ by its inverse. In these experiments, we set $\psi = 0.001$ and $w = 0.1$. Unlike in Section 7.1, we plot wall-clock time on the x-axis since everything happens in real-time but the time elapsed per round is not consistent across all algorithms.

**Baselines:** These experiments compare CONGO B, Z, and E against the same NSGD and SGDSP baselines used in Section 7.1 along with PPO as a representative pretrained RL method. We include PPO to test whether our algorithm can compete with a pretrained model since most intelligent autoscaling methods rely on pretraining (see Appendix A). Results are averaged over four runs.

**Results:** As seen in Fig. 4, CONGO-E always performs better than the gradient descent baselines, and it is able to make progress towards the optimal cost given by PPO, reducing its cost by close to 45% on average for the fixed workload and variable job type cases and 20% for the variable arrival rate case in a matter of just over two hours. Furthermore, CONGO-E reaches a lower cost faster than CONGO-Z for the FW and VAR cases, demonstrating the superiority of the Gaussian distribution for measurement matrices. On the other hand, CONGO-B shows poorer performance though it still maintains a lower cost than the gradient descent baselines at most times; we attribute this to the extra noise which the algorithm suffers from as explained in Section 4.

## 8    CONCLUSION AND LIMITATIONS

In this paper, we introduced a framework that brings the benefits from compressive sensing into the OCO setting. We proved that in the sparse gradient setting, a number of samples only logarithmic in $d$ is sufficient to guarantee $\mathcal{O}(\sqrt{T})$ regret. Furthermore, we showed that these benefits extend to practical applications. Overall, we found that CONGO-Z allows for strong theoretical guarantees with relatively few samples while CONGO-B offers poor theoretical guarantees but performs better in scenarios where observations are highly noisy, and CONGO-E provides the best of both worlds.

**Limitations:** The assumption of exactly sparse gradients in our theoretical analysis may not hold in complex systems, but our experiments in Section 7 show that near-sparsity suffices for the benefits of compressive sensing to be seen. As noted in Section 5, our algorithm assumes that bounds on several quantities are known; however, it is acceptable for the bounds on $L_f$ and $L$ to be quite loose if we are mainly concerned about the scaling of $T$ and $d$. Our treatment of the Jackson network optimization problem (and hence the microservice autoscaling problem) as an OCO problem depends upon the assumption of stability (i.e., job arrival rates do not exceed service rates for prolonged periods), and in our experiments we took measures to ensure that stability was not violated in a way that prevented the algorithms from recovering. Measures to ensure stability are detailed in Appendix G.

ACKNOWLEDGEMENTS

Research was funded in part by NSF grants CNS 2312978 and CISE Expeditions in Computing CCF-2326576. All opinions expressed are those of the authors, and need not represent those of the sponsoring agencies.

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

## A    RELATED WORKS

**Related Work: *Zeroth-Order Online Convex Optimization:*** *Online Convex Optimization* (OCO) is a game in which an online player iteratively makes decisions drawn from a convex set. Upon committing to a decision, the player incurs a loss, which is a possibly advesarially chosen convex function of the possible decisions and is unknown beforehand. The performance metric for this problem is typically the static regret based on the best action in hindsight, see Hazan et al. (2016) for a good introduction to the topic. In this setting, there are different levels of information that a learning algorithm may have access to. The first is full information, where the full objective function chosen by the adversary is revealed after the corresponding cost is incurred. If the objective function is also smooth, then regret which scales as $\sqrt{T}$ is achievable by simply running a gradient descent algorithm on the gradients of the revealed functions. Another level of information is bandit information (Dani et al., 2007; Saha and Tewari, 2011), where only the loss is revealed; in this case it is common to rely on zeroth-order methods for gradient estimation. Several works have not only been able to characterize information theoretic minimax regret bounds under bandit feedback (see for example Lattimore (2020); Bubeck et al. (2015)) but have also provided algorithms that can achieve these bounds ( Flaxman et al. (2004); Balasubramanian and Ghadimi (2018); Ito (2020); Saha and Tewari (2011); Lattimore and György (2023) to name a few). An intermediate setting between these two extremes allows the algorithm to collect a limited number of samples from the objective function after incurring the loss, which opens up more options for zeroth-order gradient estimation procedures. Our work belongs to this setting. A general oracle framework for zeroth-order gradient estimation procedures using one or more samples in the OCO context is developed in (Hu et al., 2020).

***Simultaneous Perturbation Stochastic Approximation:*** The most basic form of SPSA, due to (Spall, 1992), is a method for computing an estimate of $\nabla f(\boldsymbol{x})$ using only zeroth-order information with only two evaluations of the objective function regardless of the dimension of $\boldsymbol{x}$. Naively, one may compute an approximate measurement of the gradient at $\boldsymbol{x}$ along one dimension using the finite difference:

$$\frac{\partial f(\boldsymbol{x})}{\partial x_i} \approx \frac{f(\boldsymbol{x} + \delta \boldsymbol{e}_i) - f(\boldsymbol{x})}{\delta}. \tag{9}$$

Where $\boldsymbol{e}_i$ is the unit vector in direction $i$, and $\delta$ is the smoothing parameter which controls the bias. However, for a high-dimensional problem this requires at least $d$ measurements in addition to $f(\boldsymbol{x})$ to estimate the full gradient which might be infeasible due to the time or effort required to take a measurement. The alternative provided by SPSA is as follows: First, generate a sequence of $d$ i.i.d. Rademacher random variables, $\{\Delta_j\}_{j=1}^d$. Next, generate, a perturbation vector $\Delta := \sum_{j=1}^d \boldsymbol{e}_j \Delta_j$. Finally, we may compute our gradient estimate as follows:

$$\frac{f(\boldsymbol{x} + \delta \Delta) - f(\boldsymbol{x})}{\delta \Delta_j} \approx \frac{\partial f(\boldsymbol{x})}{\partial x_i} + \sum_{i \neq j} \frac{\partial f(\boldsymbol{x})}{\partial x_i} \frac{\Delta_i}{\Delta_j}$$

When the expectation is taken over the randomness in $\Delta$, the terms in the summation are eliminated and hence this method gives an approximately unbiased estimate of $\nabla f(\boldsymbol{x})$ for appropriately small $\delta$. The variance in the estimate can be reduced by repeating the process for different random draws of $\Delta$ and averaging over the results. The SPSA method provides a way to handle high-dimensional problems when the number of available measurements is limited, but it does not directly exploit the

sparsity of $\nabla f(\boldsymbol{x})$. For further details about the qualities of the SPSA method, see Bhatnagar et al. (2013).

***Compressive Sensing:*** Results in compressive sensing (Foucart and Rauhut, 2013; Candès, 2008) state that given measurements of the form $\boldsymbol{y} = \mathbf{A}\boldsymbol{x} + \boldsymbol{e}$ where $\boldsymbol{x}$ is a sparse vector, $\mathbf{A}$ is an $m \times d$ matrix (called the *measurement matrix*) satisfying the *restricted isometry property* (RIP) appropriate to the sparsity of $\boldsymbol{x}$, and $\boldsymbol{e}$ is bounded measurement noise, an approximation $\hat{\boldsymbol{x}}$ can be obtained through various recovery methods such that $||\hat{\boldsymbol{x}} - \boldsymbol{x}||_2$ is bounded by a quantity proportional to $||\boldsymbol{e}||_2$. In this context, $m$ is called the number of measurements; in most practical situations one wants $m$ to be as small as possible. The recovery step can be formulated as the nonconvex optimization problem

$$\min_{\boldsymbol{x}} ||\boldsymbol{x}||_0 \text{ subj. to } ||\mathbf{A}\boldsymbol{x} - \boldsymbol{y}||_2 \leq ||\boldsymbol{e}||_2 \tag{10}$$

which must be relaxed or approximated in practice.

For a measurement matrix $\mathbf{A}$ and integer $s$ one may define the $k^{\text{th}}$ restricted isometry constant (RIC) $\delta_k(\mathbf{A})$ that preserves the RIP property for any $k$-sparse vector. Concretely, for any $k-$sparse vector $\boldsymbol{x}$, we need to ensure

$$(1 - \delta_k)\|\boldsymbol{x}\|_2^2 \leq \|\mathbf{A}\boldsymbol{x}\|_2^2 \leq (1 + \delta_k)\|\boldsymbol{x}\|_2^2$$

Loosely speaking, we say that $\mathbf{A}$ satisfies the RIP for a given $k$ if $\delta_k$ is small (how small we need it depends on the use case). It turns out that a measurement matrix with a sufficiently small $cs^{\text{th}}$ RIC with $c$ being a small scalar (dependent upon the recovery method) will allow the recovery of any $s-$sparse vector. Furthermore, it is known that increasing $m$ increases the likelihood with which $\mathbf{A}$ will satisfy the RIP for a given RIC. However, current guarantees for deterministic measurement matrices require $m$ to scale as $\mathcal{O}(s^2)$ while for random matrices a superior $\mathcal{O}(s \log\left(\frac{ed}{s}\right))$ scaling is achievable (though there will necessarily be a probability of failure) (Foucart and Rauhut, 2013). The available recovery methods can be broadly grouped into those based on greedy algorithms and those based on convex relaxations of equation 10 (Hosny et al., 2023). One example of a greedy algorithm is Compressive Sampling Matching Pursuit (CoSaMP) from Needell and Tropp (2008), and one example of a convex relaxation is basis pursuit (first introduced by Chen et al. (1998)). The general approach of CoSaMP is to iteratively construct a sparse vector which is as close as possible to the target vector; the goodness of the estimate on each iteration is estimated using least squares. Basis pursuit, on the other hand, replaces the 0-norm in equation 10 with the 1-norm which is convex. Solutions to the convex optimization problem can then be approximated using various possible solvers.

***Regularized Gradient Estimation:*** In the zeroth-order stochastic optimization setting, where noisy samples from a static convex objective function are used to iteratively approach the function's minimum value, prior works have considered gradient estimation techniques which exploit sparsity in the objective function gradient. Among these is Wang et al. (2018) which uses the Lasso method to recover the gradient from a set of measurements taken in the same fashion as in SPSA. An important distinction between this technique and other gradient estimators which take multiple samples is that the value of the objective function at the current control is not used by the estimator, but rather it is estimated from the data. Lemma 1 in that paper indicates that the sample complexity required to achieve a bound on the gradient estimation error is on the order of $s^2$, while the compressive sensing schemes underlying CONGO-Z and CONGO-E only require a number of samples linear in $s$. This suggests that the Lasso method is less sample efficient (at least in the noiseless case which is more commonly analyzed in the OCO setting). We have not seen any work up to this point which has attempted to apply the same technique to the OCO setting.

***Microservice Autoscaling:*** Prior approaches have tried to use several variants of learned models as controllers for the task of allocating resources to microservices. At a high-level, all prior works either rely on heavy complex training regimes, which can be infeasible in practice or make simplifying assumptions as they do not exploit the sparsity in gradients. We briefly summarize several such approaches here.

Many prior works (Qiu et al., 2020; Zhang et al., 2021; Luo et al., 2022) rely on complex and heavy offline training mechanisms. This is especially challenging in the microservice setting where frequent re-training might be necessary because applications are regularly updated leading to changes in microservice behavior and even application graphs (Huye et al., 2023).

FIRM (Qiu et al., 2020) first uses an SVM model to identify the critical bottlenecked service, and then uses an offline-trained DDPG-based model to change the resources for the critical service. A

key drawback of this technique is that training FIRM requires anomalies to be injected by artificially inducing resource bottlenecks in the system - which can be either impossible or too expensive in many cases. Injecting resource constraints might be infeasible if the application is run on a third-party cloud provider. On the other hand, it can be expensive if the application undergoes frequent changes/updates.

Sinan (Zhang et al., 2021) predicts if a given resource allocation set for a microservice application can lead to latency violations. Using the predictions, it then relies on a heuristic to choose the next control action. Sinan has two drawbacks. First, training the latency violation predictor requires extensive data on various resource allocation and workload combinations, which is again impractical. Second, the heuristic to change resource allocations might not be general enough for all applications.

Erms (Luo et al., 2022) approximates the latency-throughput curve for each microservice as a piecewise linear function and employs techniques to prioritize latency-critical services at shared microservices. The assumption of piecewise linear function can be too simplistic for most microservices. Especially, at the inflection points, when a small change in the request rate causes high changes in the latencies, even small error in the piecewise linear estimation can lead to drastically bad performance.

Autothrottle (Wang et al., 2024) uses an online RL mechanism using contextual bandits and thus, alleviates the problem of offline training to a great extent. However, because Autothrottle does not exploit the sparsity in gradients of microservices, it is forced to make a simplifying assumption by clustering all services into just two clusters, leading to the same action for all services in the same class. This would imply that even if one microservice is the bottleneck, in expectation, it could lead to more resources being allocated for roughly half of all the services!

Erlang (Sachidananda and Sivaraman, 2024) is another such online RL-based system for microservice autoscaling, that uses a multi-armed bandit algorithm to choose the right resource allocation, given a congested microservice. However, Erlang relies on a simple heuristic to choose the congested microservice in the first place. The heuristic is to just use the service which has the highest change in CPU utilization. This heuristic may not hold in many cases, for example, when the CPU utilization of a microservice may be high because it is continuously polling on a downstream microservice (Gan et al., 2019).

## B   DESCRIPTION OF CONGO-B ALGORITHM

In this section we give details for the CONGO-B algorithm, including the pseudocode in Algorithm 2. CONGO-B takes inspiration from Algorithm 1 in Borkar et al. (2018). The functional differences between that algorithm and the gradient estimation procedure used in CONGO-B are summarized below:

- We scale the perturbation by $||\mathbf{A}^{t^T}\Delta||_2^{-2}$
- We rescale $\mathbf{A}^t$ and $\boldsymbol{y}$ by $\frac{1}{\sqrt{m}}$ right before the recovery step
- We introduce an additional constraint to the basis pursuit $l1$-minimization problem to ensure a specific bound on the gradient estimates used for gradient descent steps

We emphasize that the inclusion of the additional constraint does not affect the applicability of the recovery result used in the proof of Lemma 2. To see this, note that under any outcome where the solution to the $l1$-minimization problem without the second constraint satisfies $||\tilde{\nabla} f(\boldsymbol{x}) - \nabla f(\boldsymbol{x})||_2 \leq 3L\delta$, we also have $||\tilde{\nabla} f_t(\boldsymbol{x}_t)||_2 \leq 3L\delta + ||\nabla f_t(\boldsymbol{x}_t)||_2 \leq 3L\delta + L_f$ by the triangle inequality, and hence the second constraint is immediately satisfied.

---

**Algorithm 2** CONGO-B: Compressive Online Gradient Optimization Borkar Version

---

**Require:** Constraint set $\mathcal{K}$, Horizon $T$, Lipschitz constant $L_f$, Smoothness parameter $L$, Parameters $(\eta_t)_{t=1}^T$, $\delta$, $m$, Initialization point $x_1$

    **for** $t = 1, 2, \cdots, T$ **do**
        $\mathbf{A}^t \leftarrow$ randomly drawn $m \times d$ matrix with $\mathcal{N}(0,1)$ entries     ▷ *Generate measurement matrix*
        Query $f_t(\boldsymbol{x}_t)$
        **for** $l = 1, 2, \cdots, k$ **do**
            $\Delta^l \leftarrow$ random $m$-dimensional vector of Rademacher random variables
            Query at point $\boldsymbol{x}_t + \delta||\mathbf{A}^t||_2^{-2}\mathbf{A}^{t^T}\Delta^l$ to get $f_t(\boldsymbol{x}_t + \delta||\mathbf{A}^t||_2^{-2}\mathbf{A}^{t^T}\Delta^l)$
            $\beta^l \leftarrow f_t(\boldsymbol{x}_t + \delta||\mathbf{A}^t||_2^{-2}\mathbf{A}^{t^T}\Delta^l) - f_t(\boldsymbol{x}_t)$
            **for** $i = 1, 2, \cdots, m$ **do**
                $y_i^l \leftarrow \frac{\beta^l}{\delta||\mathbf{A}^t||_2^{-2}\Delta_i^l}$         ▷ *Construct $\boldsymbol{y}$ from measurements*
            **end for**
        **end for**
        $\boldsymbol{y} \leftarrow \left(\frac{1}{k}\sum_{l=1}^{k_t} y_1^l, \cdots, \frac{1}{k}\sum_{l=1}^{k_t} y_m^l\right)$         ▷ *Average over observations*
        Rescale $\mathbf{A}^t$ and $\boldsymbol{y}$ by $\frac{1}{\sqrt{m}}$
        **if** There exists a feasible solution $\boldsymbol{z}^*$ for         ▷ *Constrained gradient recovery*
                $\min ||\boldsymbol{z}||_1$ subj. to $||\mathbf{A}^t\boldsymbol{z} - \boldsymbol{y}||_2 \leq 3L\delta$ and $||\boldsymbol{z}||_2 \leq L_f + 3L\delta$
        **then**
            $\tilde{\nabla} f_t(\boldsymbol{x}_t) \leftarrow \boldsymbol{z}^*$
        **else**
            $\tilde{\nabla} f_t(\boldsymbol{x}_t) \leftarrow 0$
        **end if**
        $\boldsymbol{x}_{t+1} \leftarrow \underset{\boldsymbol{x} \in \mathcal{K}}{\operatorname{argmin}} ||\boldsymbol{x} - (\boldsymbol{x}_t - \eta_t \tilde{\nabla} f_t(\boldsymbol{x}_t))||_2$     ▷ *Gradient descent update*
    **end for**
    Incur regret $\sum_{t=1}^T f_t(\boldsymbol{x}_t) - f_t(\boldsymbol{x}^*)$

---

## C  PROOFS

### C.1  PROOF OF LEMMA 1

*Proof.* We begin with the case where

$$y_i = \left(f(\boldsymbol{x} + \frac{\delta}{||\mathbf{a}_i||_2^2}\mathbf{a}_i) - f(\boldsymbol{x})\right)\frac{||\mathbf{a}_i||_2^2}{\delta} \quad \forall i \in [m]$$

Applying the Taylor approximation given in equation 6 and the bound given in equation 7, the expression simplifies as follows:

$$y_i = \left\langle \nabla f(\boldsymbol{x}), \frac{\delta}{||\mathbf{a}_i||_2^2}\mathbf{a}_i\right\rangle \frac{||\mathbf{a}_i||_2^2}{\delta} + \xi \frac{||\mathbf{a}_i||_2^2}{\delta}$$
$$\leq \langle \nabla f(\boldsymbol{x}), \mathbf{a}_i\rangle + \frac{1}{2}\left(\frac{\delta}{||\mathbf{a}_i||_2^2}\right)^2 ||\mathbf{a}_i||_2^2 ||\nabla^2 f||_2 \frac{||\mathbf{a}_i||_2^2}{\delta}$$
$$\leq \langle \nabla f(\boldsymbol{x}), \mathbf{a}_i\rangle + \frac{L}{2}\delta$$

It follows that $\boldsymbol{y} = \mathbf{A}\nabla f(\boldsymbol{x}) + \mathbf{e}$ where each element of $\mathbf{e}$ is bounded by $\frac{L}{2}\delta$. Since $\mathbf{e} \in \mathbb{R}^m$, we have $||\mathbf{e}||_2 \leq \frac{L}{2}\delta\sqrt{m}$.

Now we turn to the case where

$$y_i = \left(f(\boldsymbol{x} + \frac{\delta}{||\mathbf{A}^T\Delta||_2^2}\mathbf{A}^T\Delta) - f(\boldsymbol{x})\right)\frac{||\mathbf{A}^T\Delta||_2^2}{\delta\Delta_i} \quad \forall i \in [m]$$

Simplifying as before gives:

$$
\begin{aligned}
y_i &= \left\langle \nabla f(\boldsymbol{x}), \frac{\delta}{||\mathbf{A}^T\Delta||_2^2}\mathbf{A}^T\Delta \right\rangle \frac{||\mathbf{A}^T\Delta||_2^2}{\delta\Delta_i} + \xi\frac{||\mathbf{A}^T\Delta||_2^2}{\delta\Delta_i} \\
&\leq \sum_{j=1}^m \frac{1}{\Delta_i}\langle \nabla f(\boldsymbol{x}), \mathbf{a}_j\Delta_j\rangle + \left(\frac{\delta}{||\mathbf{A}^T\Delta||_2^2}\right)^2 ||\mathbf{A}^T\Delta||_2^2\frac{L}{2}\frac{||\mathbf{A}^T\Delta||_2^2}{\delta} \\
&= \langle \nabla f(\boldsymbol{x}), \mathbf{a}_i\rangle + \sum_{j\neq i}\frac{\Delta_j}{\Delta_i}\langle \nabla f(\boldsymbol{x}), \mathbf{a}_j\rangle + \frac{L}{2}\delta
\end{aligned}
$$

If we let $[\mathbf{e}_1]_i = \sum_{j\neq i}\frac{\Delta_j\langle\nabla f(\boldsymbol{x}),\mathbf{a}_j\rangle}{\Delta_i}$ and $[\mathbf{e}_2]_i = \xi\frac{||\mathbf{A}^T\Delta||_2^2}{\delta\Delta_i}$, we have $\boldsymbol{y} = \mathbf{A}\nabla f(\boldsymbol{x}) + \mathbf{e}_1 + \mathbf{e}_2$ as desired, and as before we have $||\mathbf{e}_2||_2 \leq \frac{L}{2}\delta\sqrt{m}$. □

## C.2 PROOF OF THEOREM 1

### C.2.1 ADDITIONAL LEMMA FOR THE PROOF OF THEOREM 1

The proof of Theorem 1 requires the following lemma from (Nemirovski et al., 2009) expressed in our notation below.

**Lemma 3** (Lemma 2.1 of (Nemirovski et al., 2009)). *Let $V_\omega(\cdot, \cdot)$ be the Bregman Distance w.r.t. the $\alpha$-strongly convex distance generating function $\omega$, and let $P_x(\cdot)$ be the proximal mapping associated with $\omega$. Then for every $u \in \mathcal{K}$, $x \in \mathcal{K}^o$, and $y \in \mathbb{R}^d$, one has*

$$
V_\omega(P_x(y), u) \leq V_\omega(x, u) + \langle y, u - x\rangle + \frac{||y||_*^2}{2\alpha}
$$

*Using $x = x_t$ and $y = \eta_t\tilde{\nabla}f_t(\boldsymbol{x}_t)$, we get*

$$
\eta_t\langle x_t - u, \tilde{\nabla}f_t(\boldsymbol{x}_t)\rangle \leq V_\omega(x_t, u) - V_\omega(x_{t+1}, u) + \frac{\eta_t^2}{2\alpha}||\tilde{\nabla}f_t(\boldsymbol{x}_t)||_*^2
$$

This result applies generally to mirror descent algorithms, but since projected gradient descent can be viewed as a special case of this in the Euclidean setting, we simplify it to equation 11 (This form appears in equation 2.6 of (Nemirovski et al., 2009), where the expectation is taken over all of the terms).

$$
\langle\tilde{\nabla}f_t(\boldsymbol{x}_t), x_t - u\rangle \leq \frac{1}{\eta_t}\left(\frac{1}{2}||u - x_t||_2^2 - \frac{1}{2}||u - x_{t+1}||_2^2\right) + \frac{\eta_t}{2}||\tilde{\nabla}f_t(\boldsymbol{x}_t)||_2^2 \tag{11}
$$

### C.2.2 ASSUMPTIONS

The following must hold for all $t \in [T]$:

- $||\tilde{\nabla}f_t(\boldsymbol{x}_t) - \nabla f_t(\boldsymbol{x}_t)||_2 \leq M\delta$ with probability at least $1 - \frac{1}{T}$ for some $M$ independent of $T$
- $||\tilde{\nabla}f_t(\boldsymbol{x}_t)||_2 \leq M\delta + L_f$

### C.2.3 PROOF OF THEOREM 1

*Proof.* First, we apply the law of total expectation and the above assumptions to derive a bound on $\mathbb{E}\left[\left\|\tilde{\nabla}f_t(\boldsymbol{x}_t) - \nabla f_t(\boldsymbol{x}_t)\right\|_2\right]$. The second assumption implies that with probability 1,

$$
||\tilde{\nabla}f_t(\boldsymbol{x}_t) - \nabla f_t(\boldsymbol{x}_t)||_2 \leq M\delta + L_f + ||\nabla f_t(\boldsymbol{x}_t)||_2 \leq M\delta + 2L_f
$$

Let $E$ denote the event $||\tilde{\nabla}f_t(\boldsymbol{x}_t) - \nabla f_t(\boldsymbol{x}_t)||_2 \leq M\delta$ and $E^c$ its complement. By the law of total expectation,

$$
\begin{aligned}
\mathbb{E}\left[\left\|\tilde{\nabla}f_t(\boldsymbol{x}_t) - \nabla f_t(\boldsymbol{x}_t)\right\|_2\right] &= \mathbb{E}\left[\left\|\tilde{\nabla}f_t(\boldsymbol{x}_t) - \nabla f_t(\boldsymbol{x}_t)\right\|_2 \,\Big|\, E\right]\mathbb{P}(E) \\
&\quad + \mathbb{E}\left[\left\|\tilde{\nabla}f_t(\boldsymbol{x}_t) - \nabla f_t(\boldsymbol{x}_t)\right\|_2 \,\Big|\, E^c\right]\mathbb{P}(E^c) \\
&\leq M\delta + (M\delta + 2L_f)\frac{1}{T}
\end{aligned}
$$

where we use the bound $\mathbb{P}(E) \leq 1$.

Now, informed by the approach in Section 7.2 of (Hu et al., 2020), we seek to show that on each round the regret of our algorithm compared to any arbitrary point, which includes $\min_{\boldsymbol{x}^*} \sum_{t=0}^{T-1} f_t(\boldsymbol{x}^*)$, is sufficiently small. We begin by applying the convexity of $f_t$:

$$
\begin{aligned}
\mathbb{E}\left[\sum_{t=1}^{T} f_t(x_t) - \sum_{t=1}^{T} f_t(x)\right] &\leq \mathbb{E}\left[\sum_{t=1}^{T} \langle \nabla f_t(\boldsymbol{x}_t), x_t - x\rangle\right] \\
&= \mathbb{E}\left[\sum_{t=1}^{T} \langle \tilde{\nabla} f_t(\boldsymbol{x}_t), x_t - x\rangle + \sum_{t=1}^{T} \langle \nabla f_t(\boldsymbol{x}_t) - \tilde{\nabla} f_t(\boldsymbol{x}_t), x_t - x\rangle\right] \\
&\leq \mathbb{E}\left[\sum_{t=1}^{T} \langle \tilde{\nabla} f_t(\boldsymbol{x}_t), x_t - x\rangle\right] + \sum_{t=1}^{T} \mathbb{E}\left[\left\|\nabla f_t(\boldsymbol{x}_t) - \tilde{\nabla} f_t(\boldsymbol{x}_t)\right\|_2 \|x_t - x\|_2\right] \\
&\leq \mathbb{E}\left[\sum_{t=1}^{T} \langle \tilde{\nabla} f_t(\boldsymbol{x}_t), x_t - x\rangle\right] + \sum_{t=1}^{T} \mathbb{E}\left[2R \left\|\nabla f_t(\boldsymbol{x}_t) - \tilde{\nabla} f_t(\boldsymbol{x}_t)\right\|_2\right] \\
&\leq \mathbb{E}\left[\sum_{t=1}^{T} \langle \tilde{\nabla} f_t(\boldsymbol{x}_t), x_t - x\rangle\right] + 2RT\left(M\delta + (M\delta + 2L_f)\frac{1}{T}\right)
\end{aligned}
$$

Focusing on the remaining term, we apply Lemma 3 in the form given by equation 11, where we let $u = x$. Thus, by telescoping,

$$
\begin{aligned}
\mathbb{E}\left[\sum_{t=1}^{T} \langle \tilde{\nabla} f_t(\boldsymbol{x}_t), x_t - x\rangle\right] &\leq \mathbb{E}\left[\sum_{t=1}^{T} \frac{1}{\eta_t}\left(\frac{1}{2}\|x - x_t\|_2^2 - \frac{1}{2}\|x - x_{t+1}\|_2^2\right) + \eta_t \mathbb{E}\left[\frac{\|\tilde{\nabla} f_t(\boldsymbol{x}_t)\|_2^2}{2}\right]\right] \\
&\leq \frac{D}{\eta_{T-1}} + \mathbb{E}\left[\sum_{t=1}^{T} \eta_t \frac{\|\tilde{\nabla} f_t(\boldsymbol{x}_t)\|_2^2}{2}\right]
\end{aligned}
$$

where $D = \sup_{x,y \in \mathcal{K}} \frac{1}{2}\|x - y\|_2^2$. Now, we are assuming that $\|\tilde{\nabla} f_t(\boldsymbol{x}_t)\|_2 \leq M\delta + L_f$, and hence

$$
\mathbb{E}\left[\sum_{t=1}^{T} \langle \tilde{\nabla} f_t(\boldsymbol{x}_t), x_t - x\rangle\right] \leq \frac{D}{\eta_{n-1}} + \sum_{t=1}^{T} \eta_t \frac{R(M\delta + L_f)^2}{2}
$$

Putting things together, we have

$$
\begin{aligned}
\mathbb{E}\left[\sum_{t=1}^{T} f_t(x_t)\right] - \sum_{t=1}^{T} f_t(x) &\leq \frac{D}{\eta_{n-1}} + \sum_{t=1}^{T} \eta_t \frac{R(M\delta + L_f)^2}{2} \\
&\quad + 2RT\left(M\delta + (M\delta + 2L_f)\frac{1}{T}\right)
\end{aligned}
\tag{12}
$$

Plugging in $\eta_t = \frac{1}{L_f \sqrt{T}}$ and choosing $\delta = \frac{1}{MT}$ gives us

$$
\begin{aligned}
\mathbb{E}\left[\sum_{t=1}^{T} f_t(xs_t)\right] - \sum_{t=1}^{T} f_t(x) &\leq DL_f\sqrt{T} + \sqrt{T}\frac{R(\frac{1}{T} + L_f)^2}{2L_f} + 2R + \frac{2R}{T} + 4RL_f \\
&= DL_f\sqrt{T} + \frac{R}{2L_f T^{\frac{3}{2}}} + \frac{R}{\sqrt{T}} + \frac{RL_f\sqrt{T}}{2} + 2R + \frac{2R}{T} + 4RL_f
\end{aligned}
$$

For large $T$, the terms which dominate are those which scale as $\sqrt{T}$. Isolating those terms gives us the asymptotic regret of $\mathcal{O}((D + \frac{R}{2})L_f\sqrt{T})$. □

## C.3 PROOFS OF THEOREM 2 AND THEOREM 3

### C.3.1 PROOF OF LEMMA 2

*Proof.* All of the existing results used in this proof come from Foucart and Rauhut (2013). We use several of their results to determine how large $m$ must be to guarantee recovery with sufficiently

high probability. In this context, for the measurement matrix $\mathbf{A}$ and integer $k$ one may define the $k^{\text{th}}$ restricted isometry constant $\delta_k(\mathbf{A})$ as the smallest such value for which $\mathbf{A}$ satisfies the RIP property. Concretely, for any $k-$sparse vector $\boldsymbol{x}$, we need to ensure

$$(1 - \delta_k)\|x\|_2^2 \leq \|Ax\|_2^2 \leq (1 + \delta_k)\|x\|_2^2$$

Indeed, Corollary 9.3, which applies to both Gaussian and Bernoulli random matrices, gives a lower bound on the probability with which the matrix will satisfy the $\delta_k$-RIP for some integer $k$:

$$\mathbb{P}\left(\delta_k < \delta\right) \geq 1 - \epsilon \text{ if } m \geq C'\delta^{-2}\left(k\log\left(\frac{ed}{k}\right) + \log(2\epsilon^{-1})\right)$$

Note that the values of $k$ and $\delta$ required by different recovery methods differ. We will be interested in $k = 2s$ for basis pursuit and $k = 4s$ for CoSaMP. Theorem 6.12 states that it is sufficient to have $\delta_{2s} < 0.6246$ when basis pursuit is used, while Theorem 6.27 states that it is sufficient to have $\delta_{4s} < 0.4782$ when CoSaMP is used. (This requirement has been improved upon more recently (Zhao and Luo, 2020), but we do not focus on optimizing constants here). In the case of basis pursuit, when the RIP holds we get the following bound on the recovery error:

$$||\tilde{\nabla}f_t(\boldsymbol{x}_t) - \nabla f_t(\boldsymbol{x}_t)||_2 \leq \frac{C_1^{\text{BP}}}{\sqrt{s}}||\nabla f_t(\boldsymbol{x}_t)_{\bar{S}}||_1 + C_2^{\text{BP}}\frac{1}{\sqrt{m}}||\boldsymbol{e}||_2$$

where $\boldsymbol{e} = \boldsymbol{y} - \mathbf{A}^t\nabla f_t(\boldsymbol{x}_t)$ is the measurement error *before* rescaling by $\frac{1}{\sqrt{m}}$, and $C_1^{\text{BP}}$ and $C_2^{\text{BP}}$ are constants depending only on $\delta_{2s}$. In the case of CoSaMP, we instead get the following bound:

$$||\tilde{\nabla}f_t(\boldsymbol{x}_t) - \nabla f_t(\boldsymbol{x}_t)||_2 \leq \rho^n||\boldsymbol{x}_0 - \nabla f_t(\boldsymbol{x}_t)_S||_2 + C^{\text{CoSaMP}}\frac{1}{\sqrt{m}}||\mathbf{A}^t\nabla f_t(\boldsymbol{x}_t)_{\bar{S}} + \boldsymbol{e}||_2$$

where $\rho < 1$, $n$ is the number of iterations of the CoSaMP algorithm, and $C^{\text{CoSaMP}}$ is a constant depending only on $\delta_{4s}$.

Under the assumption that $\nabla f_t(\boldsymbol{x}_t)$ is $s$-sparse, $\nabla f_t(\boldsymbol{x}_t)_S = \nabla f_t(\boldsymbol{x}_t)$ and $\nabla f_t(\boldsymbol{x}_t)_{\bar{S}} = 0$ so the bound for basis pursuit becomes

$$||\tilde{\nabla}f_t(\boldsymbol{x}_t) - \nabla f_t(\boldsymbol{x}_t)||_2 \leq C_2^{\text{BP}}\frac{1}{\sqrt{m}}||\boldsymbol{e}||_2$$

Furthermore, if we assume $n$ is sufficiently large that the $\rho^n||\nabla f_t(\boldsymbol{x}_t)||_2$ term vanishes on every round, then the bound for CoSaMP becomes

$$||\tilde{\nabla}f_t(\boldsymbol{x}_t) - \nabla f_t(\boldsymbol{x}_t)||_2 \leq C^{\text{CoSaMP}}\frac{1}{\sqrt{m}}||\boldsymbol{e}||_2$$

Thus in this specific case, both basis pursuit and CoSaMP give a recovery error proportional to $||\boldsymbol{e}||_2$. For CONGO-Z and CONGO-E to be implementable in practice, it is necessary to give bounds on $C_2^{\text{BP}}$ and $C^{\text{CoSaMP}}$. From the proofs of Theorems 4.22 and 6.13 we can deduce that

$$C_2^{\text{BP}} \leq 4\frac{\left(\sqrt{1 - \delta_{2s}^2} - \frac{\delta_{2s}}{4}\right)\sqrt{1 + \delta_{2s}}}{\left(4\sqrt{1 - \delta_{2s}^2} - \frac{\delta_{2s}}{4} - \delta_{2s}\right)\left(\sqrt{1 - \delta_{2s}} - \frac{\delta_{2s}}{4}\right)}$$

One can easily verify that $\delta_{2s} < 0.6246$ implies $C_2^{\text{BP}} < 3$. For CoSaMP, from the proof of Theorem 6.27 we find that

$$C^{\text{CoSaMP}} = \sqrt{\frac{2(1 + 3\delta_{4s}^2)}{1 - \delta_{4s}}} + \frac{2\sqrt{1 + \delta_{4s}}}{1 - \delta_{4s}}$$

and one can easily verify that $\delta_{4s} < 0.4782$ implies $C^{\text{CoSaMP}} < 7.21$. $\qquad\square$

**Remark 2.** *According to Section 9.3 of Foucart and Rauhut (2013), for Gaussian matrices the best known lower bound for $C'$ that allows these results to hold generally is around 80; but when $\frac{d}{s}$ is sufficiently large this can be improved considerably. Much like Cai et al. (2022), we ignored this large theoretical constant in our experiments and still achieved good performance.*

### C.3.2   PROOF OF THEOREM 2

*Proof.* We first find $M$ such that the assumptions for Theorem 1 hold for CONGO-B. Our analysis begins in a similar fashion to the proof of Theorem 4 in Borkar et al. (2018), but due to our modifications to the gradient estimation procedure and the recovery guarantee from compressive

sensing used we are able to obtain a different bound that is suitable for our purposes. By Lemma 1, the vector $\boldsymbol{y}$ of measurements collected on round $t$ satisfies

$$\boldsymbol{y} = \langle \nabla f_t(\boldsymbol{x}_t), \mathbf{a}_i^t \rangle + \frac{1}{k_t} \sum_{l=1}^{k_t} (\mathbf{e}_1^l + \mathbf{e}_2^l)$$

where $[\mathbf{e}_1^l]_i = \sum_{j \neq i} \frac{\Delta_j^l \langle \nabla f_t(\boldsymbol{x}_t), \mathbf{a}_j \rangle}{\Delta_i^l}$ and $||\mathbf{e}_2^l||_2 \leq \frac{L}{2} \delta \sqrt{m} \, \forall l \in [k_t]$ To bound the overall error, we will use the averaging over $k_t$ to control the magnitude of $\mathbf{e}_1$ and ignore the averaging on $\mathbf{e}_2$ since we can use $\delta$ to control its magnitude instead. Define $\gamma := \frac{L}{2} \delta \sqrt{m}$. We wish to set $k_t$ such that

$$\mathbb{P}\left( \left| \frac{1}{k_t} \sum_{l=1}^{k_t} \left( \sum_{j \neq i} \frac{\Delta_j^l \langle \nabla f_t(\boldsymbol{x}_t), \mathbf{a}_j^t \rangle}{\Delta_i^l} \right) \right| > \frac{\gamma}{\sqrt{m}} \right) \leq \frac{1}{2mT} \tag{13}$$

which, via a union bound, will ensure that $||\mathbf{e}_1 + \mathbf{e}_2||_2 \leq 2\gamma$ with probability at least $1 - \frac{1}{2T}$. We apply Hoeffding's inequality, noting that $\mathbb{E}\left[ \frac{\Delta_j^l \langle \nabla f_t(\boldsymbol{x}_t), \mathbf{a}_j^t \rangle}{\Delta_i^l} \right] = 0$ and $\left| \sum_{j \neq i} \frac{\Delta_j^l \langle \nabla f_t(\boldsymbol{x}_t), \mathbf{a}_j^t \rangle}{\Delta_i^l} \right| \leq (m-1) \max_i \langle \nabla f_t(\boldsymbol{x}_t), \mathbf{a}_i^t \rangle \leq (m-1) L_f G$, where we introduce the notation $G = \max_i ||\mathbf{a}_i^t||_2$. This gives us the following:

$$\mathbb{P}\left( \left| \frac{1}{k_t} \sum_{l=1}^{k_t} \left( \sum_{j \neq i} \frac{\Delta_j^l \langle \nabla f_t(\boldsymbol{x}_t), \mathbf{a}_j^t \rangle}{\Delta_i^l} \right) \right| > \frac{\gamma}{\sqrt{m}} \right) \leq \exp\left( \frac{-2k_t \gamma^2}{m(m-1)^2 (L_f)^2 G^2} \right)$$

Thus we must solve the following to find the desired value for $k_t$:

$$\exp\left( \frac{-2k_t \gamma^2}{m(m-1)^2 (L_f)^2 G^2} \right) \leq \frac{1}{2mT}$$

The solution is

$$k_t \geq \frac{m(m-1)^2 (L_f)^2 G^2}{2\gamma^2} \log(2mT)$$

$$= \frac{4(m-1)^2 (L_f)^2 G^2}{L^2 \delta^2} \log(2mT) \tag{14}$$

We round up this value to ensure $k_t$ is an integer. We can now apply Lemma 2, which gives us the following guarantee:

$$||\tilde{\nabla} f_t(\boldsymbol{x}_t) - \nabla f_t(\boldsymbol{x}_t)||_2 \leq 6\gamma = 3L\delta$$

which holds with probability at least $1 - (\frac{1}{2T} + \epsilon)$ as long as

$$m \geq C'(0.6246)^{-2} \left( s \log\left( \frac{ed}{s} \right) + \log(2\epsilon^{-1}) \right)$$

Setting $\epsilon = \frac{1}{2T}$ ensures that assumption 1 of Theorem 1 is satisfied. Assumption 2 is also satisfied because we replace $\tilde{\nabla} f_t(\boldsymbol{x}_t)$ with the zero vector when it exceeds $3L\delta + L_f$. The desired bound on the expected regret follows from Theorem 1 with $M = 3L$.

As a requirement of Theorem 1, we must set $\delta = \frac{1}{MT} = \frac{1}{3LT}$ Note that $G$ found in the expression for $k_t$ is a random variable which is only known once $\mathbf{A}^t$ is drawn, hence the reason why the number of samples to average over varies for different rounds. we finish by bounding the expected value of $k_t$ (which is the same on every round since $\mathbf{A}^t$ is drawn the same way on every round). To do so, we

must bound $\mathbb{E}\left[G^2\right] = \mathbb{E}\left[\max_i \|\mathbf{a}_i^t\|_2^2\right]$. Let $Y = \max_i \|\mathbf{a}_i^t\|_2^2$, and observe that

$$\exp\left(t\mathbb{E}\left[Y\right]\right) \leq \mathbb{E}\left[\exp\left(tY\right)\right]$$

$$\leq \mathbb{E}\left[\exp\left(t\sum_{i=1}^m \|\mathbf{a}_i^t\|_2^2\right)\right]$$

$$\leq \mathbb{E}\left[\sum_{i=1}^m \exp\left(t\|\mathbf{a}_i^t\|_2^2\right)\right]$$

$$= \sum_{i=1}^m \mathbb{E}\left[\exp\left(t\|\mathbf{a}_i^t\|_2^2\right)\right]$$

$$= m(1 - 2t)^{-\frac{d}{2}} \ \forall t \in (0, 0.5)$$

Where we apply the moment generating function of a $\chi^2(d)$ distribution. Taking the logarithm of both sides and dividing by $t$ gives us

$$\mathbb{E}\left[Y\right] \leq \frac{d}{2t}\log\left(\frac{m}{1-2t}\right)$$

Choosing $t = \frac{1}{4}$ yields

$$\mathbb{E}\left[G^2\right] = \mathbb{E}\left[Y\right] \leq 2d\log(2m)$$

Plugging these values into the expected value of $k_t$ as

$$\mathbb{E}\left[k_t\right] \leq 36d(m-1)^2 L_f^2 \log(2m)T\log(2mT)$$

$\square$

### C.3.3 Proof of Theorem 3

*Proof.* As in the proof of Theorem 2, we first find $M$ such that the assumptions for Theorem 1 hold. Lemma 1 tells us that the error we must account for in the recovery process is bounded by $e \leq \frac{L}{2}\delta\sqrt{m}$, which after rescaling becomes $\frac{L}{2}\delta$. Lemma 2 tells us that $M = \frac{7.21}{2}L$ is sufficient if we have

$$m \geq C'(0.4782)^{-2}\left(s\log\left(\frac{ed}{s}\right) + \log(2\epsilon^{-1})\right)$$

Note that the value of $C'$ is different for CONGO-Z and CONGO-E due to the different measurement matrices (this is in fact the only difference in the proofs for these algorithms). We set $\epsilon = \frac{1}{T}$ to satisfy Assumption 1. Assumption 2 is also satisfied because we replace $\tilde{\nabla}f_t(\boldsymbol{x}_t)$ with the zero vector when it exceeds $\frac{7.21}{2}L\delta + L_f$ (respectively $\frac{3}{2}L\delta + L_f$ in the case of basis pursuit). Applying Theorem 1 finishes the proof. $\square$

## D MOVING FROM STOCHASTIC OPTIMIZATION TO ONLINE CONVEX OPTIMIZATION

Here we further discuss the rationale behind some elements of the CONGO framework. We mention in Section 4 that it is necessary to enforce a limit on the magnitude of any gradient estimate that is applied as a gradient descent step. This requirement arises from the use of Lemma 3 when we bound the cumulative regret in Theorem 1, which introduces a term involving $\|\tilde{\nabla}f_t(\boldsymbol{x}_t)\|_2^2$. Without a bound on the magnitude of the gradient estimates that holds with probability one, it becomes difficult to deal with this term in the regret bound. Note that an analysis like the one for stochastic optimization setting in Cai et al. (2022) which yields a high probability bound by conditioning on the measurement matrix being "good" does not have to worry about issues like this because it can rely on a recovery guarantee holding. Now, if we know a bound $L_f$ on the norm of the true gradient, then it makes sense to enforce a bound on the estimated gradient equal to $L_f$ plus the maximum error we are willing to tolerate, since we know that any estimate which does not satisfy this bound is not useful to us. Since

this is a rare event (as proven in Lemma 2), we can afford to skip the gradient descent step to avoid making a very large step which may be in a bad direction. An example of this in practice is given in Fig. 7. Note that as a consequence, it is necessary to know a bound on $L_f$ in order for the theoretical guarantee to hold.

As also mentioned at the end of Section 4, under the CONGO framework we sample a new measurement matrix on each round. In the ZORO algorithm, a single measurement matrix is sampled at the start and reused for every gradient estimate. This makes sense when the goal is to obtain a high-probability sample complexity bound since one only needs to bound the probability of a single event rather than a separate event for each round. However, when the goal is to prove a bound on the expected regret as in our case, it becomes irrelevant whether the matrix is drawn once overall or on each round. This is because applying the law of total expectation on the entire cumulative regret leads to the same result as applying it to the gradient estimation error does. We choose to re-sample the measurement matrix on each round to eliminate the possibility of a single "bad" measurement matrix ruining all of the rounds, and hence we apply the law of total expectation to the gradient estimation error alone. The fact that we are pursuing a bound on the regret in expectation also influences how we choose $m$. After applying the law of total expectation, it remains necessary to bound the gradient error when compressive sensing recovery fails which necessarily introduces a term that does not scale with $\delta$; to compensate for this we need the probability of failure to decrease as $T$ increases. This is why our choice of $m$ for the theoretical results depends logarithmically on $T$ while in Cai et al. (2022) it does not. On the other hand, if one simply wants a bound on the regret which holds with a high probability independent of $T$, then the method of sampling a single measurement matrix at the start could be adopted, in which case $\epsilon$ could be set to the desired probability instead of $\frac{1}{T}$ since the magnitude of regret when the high probability event does not occur no longer matters.

## E   Practical Implementation of Recovery Methods

Here we justify our choice of the CoSaMP algorithm for CONGO-E based on practical considerations. Basis pursuit is in theory a good candidate for a recovery algorithm because it allows for fewer measurements to be used than methods like CoSaMP and Iterative Hard Thresholding (this can be seen by comparing the requirements on the restricted isometry constant for each algorithm; see Chapter 6 of Foucart and Rauhut (2013)). However, the practical implementations of basis pursuit have poor runtime guarantees compared to greedy methods like CoSaMP (Hosny et al., 2023). Empirically, we have observed that for the same tolerance level and limit on the number of iterations, the implementation of basis pursuit using Chambolle and Pock's primal-dual algorithm (see Section 15.2 of Foucart and Rauhut (2013) for the pseudocode) is indeed slower than CoSaMP. Fig. 5 compares the CDF of the execution time for the two methods; observe that the curve for the primal-dual algorithm stays to the right of the curve for CoSaMP. Furthermore, using basis pursuit for robust recovery requires one to have a bound on $||\mathbf{e}||_2$ which will hold with high probability whereas CoSaMP only requires an estimate of the sparsity level (Needell and Tropp, 2008). Based on these considerations, we rely on the CoSaMP algorithm for the implementations of CONGO-Z and CONGO-E used in our experiments. For CONGO-B, we use basis pursuit to remain faithful to the method proposed by Borkar et al. (2018). Note that both CoSaMP and basis pursuit have additional hyperparameters which determine the tolerance and maximum number of iterations to run; the exact values used are given in Appendix G.

## F   Additional Numerical Simulation Results

### F.1   Analysis of the impact of different choices for $m$

In our numerical simulation environment, we ran experiments to test how the quality of the gradient estimate from CONGO-E with CoSaMP changes as $m$ varies, and to see if the value of $2s \log \left( \frac{d}{s} \right)$ prescribed in the theoretical results (ignoring the $\log(T)$ term) is in fact a good choice in practice. First, we set $d = 50$, $s = 5$ and $R = 50$ and ran CONGO-E for each of the values of $m$ in the range [6,24] on the same environment with the same 50 random seeds. We computed $||\tilde{\nabla} f_t(\boldsymbol{x}_t) - \nabla f_t(\boldsymbol{x}_t)||_2$ on each round for each algorithm and averaged over all the measurements at the end. Fig. 6 shows a bar graph of the average errors for the different values of $m$. Note how the error rapidly drops after $m = 15$; in this case, $2s \log \left( \frac{d}{s} \right) = 23.026$ so $m$ would be 24 after rounding up. This is clearly in the

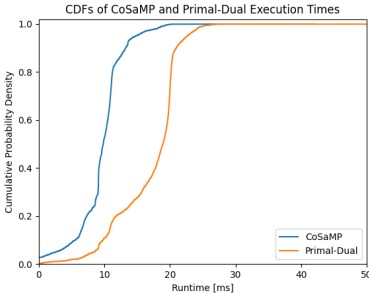

Figure 5: Comparison of the CDFs for execution time of basis pursuit and CoSaMP over 1000 executions.

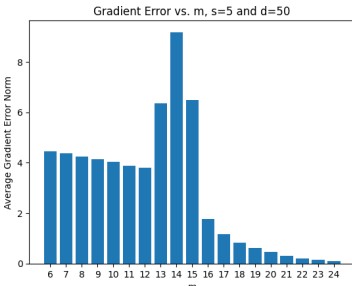

Figure 6: Bar graph showing how error in the gradient estimate changes as more samples are used.

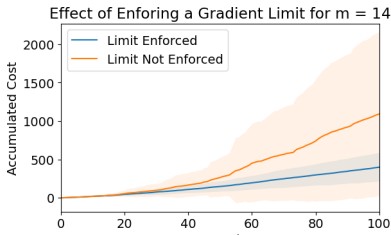

Figure 7: Comparison of the the cumulative cost with and without enforcing a limit on the gradients used for online gradient descent in the case where $m$ is insufficiently large

region where the gradient error is very small and so there is no need to increase $m$ further. Note that we also ran the GDSP algorithm under the same conditions with a total of 26 samples per gradient estimate (including the sample for $f_t(x_t)$) and found that the average norm of the gradient error was 31.64, far higher than CONGO with even just 16 samples per gradient estimate. Surprisingly, however, we observe a very sharp spike in gradient error for $m = 13$, 14, and 15. This is accompanied by abnormally large values in the gradient estimates. Further investigation found that this was not simply caused by a poorly conditioned measurement matrix, since the maximum and minimum eigenvalues of $\mathbf{A}^t$ were not outside the ordinary range when these highly erroneous gradient estimates were made. The anomaly occurs both for basis pursuit and CoSaMP. While we do not know the exact cause of this behavior, the implication is that allowing $m$ to be significantly smaller than the prescribed value runs the risk of very poor performance. Of course, due to the constraint on the magnitude of the gradient estimate, when this anomaly occurs its contribution to the cumulative regret will be limited. To illustrate this, we compared the cumulative regret for certain value of $m$ with and without enforcing the gradient limit (see Fig. 7) and observed that for $m = 14$ a significant amount of regret could be avoided.

### F.2 ANALYSIS OF ROBUSTNESS TO SPARSITY LEVEL

To demonstrate that CONGO-style algorithms (particularly CONGO-Z and CONGO-E) can still be useful in scenarios where (i) $s$ is not significantly smaller than $d$, (ii) the gradient is not exactly $s$-sparse but $s$ dimensions are much larger than the rest, or (iii) the true value of $s$ is unknown and must be guessed with some level of error, we ran additional numerical simulations for each of these cases using a setup similar to the experiments of Section 6.

In the left plot of Fig. 8, we consider the noiseless case where $d = 50$ and test on $s \in \{5, 10, 15, 20\}$. We also set $c = 4$ in equation 8 to ensure that the cumulative costs come out positive. In this experiment, we allow CONGO-B to use three times as many samples per round as the other algorithms. We plot the excess cost that each algorithm accumulates over $T = 100$ rounds compared to exact gradient descent. We find that while GDSP and CONGO-B do progressively worse compared to

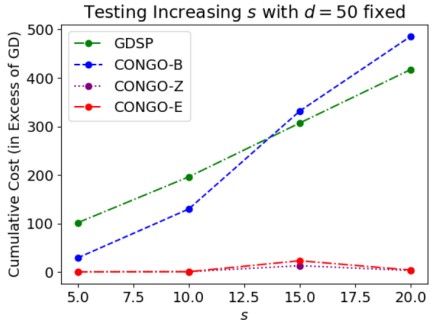 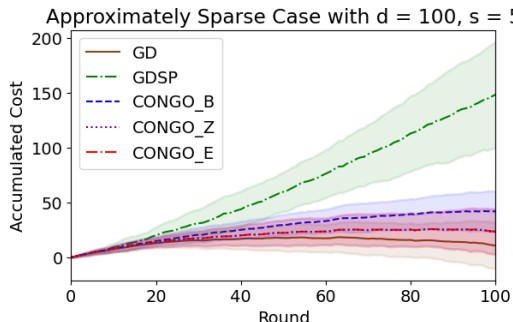

Figure 8: Left: Cumulative costs incurred beyond that of exact gradient descent over 100 rounds when $s$ increases while $d$ is fixed. The performance of CONGO-Z and CONGO-E is not significantly different from that of exact gradient descent even in low-sparsity situations. Right: Cumulative cost trajectories in a scenario where gradients are only approximately sparse. The CONGO-style algorithms continue to perform much better than GDSP.

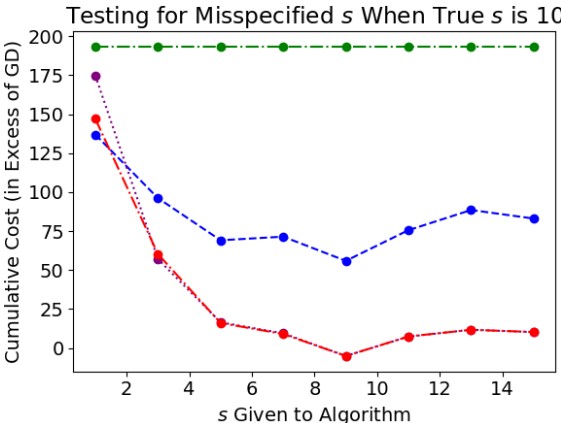

Figure 9: Cumulative costs incurred beyond that of exact gradient descent over 100 rounds when the value of $s$ assumed by the algorithm differs from the true value. CONGO-Z and CONGO-E achieve low excess cost compared to exact gradient descent even when the assumed value of $s$ is less than the true value of 10.

exact gradient descent as $s$ increases, CONGO-Z and CONGO-E remain relatively stable when using the prescribed number of samples.

In the right plot of Fig. 8, we set $d = 100$ and $s = 5$ but in this case, the elements of $\boldsymbol{b}$ and the main diagonal of $\boldsymbol{D}$ in equation 8 outside of the $s$ "significant" dimensions are scaled by $\frac{1}{d}$ rather than being set to zero. Thus the objective function on each round is only approximately sparse. Here we set $c = 2$ and allow CONGO-B to use $3m$ samples per round as before. One can see that although the performance of CONGO-Z and CONGO-E does not match that of exact gradient descent (as it does in the first experiment of Section 6), it is not too much worse. The results support our claim that the assumption of exact gradient sparsity is not required for CONGO-style algorithms to outperform algorithms which do not exploit sparsity.

In Fig. 9, we consider the case where $d = 100$ and $s = 10$ but the value of $s$ which the algorithm uses varies. In this experiment we set $c = 2$ and allow CONGO-B to use $3m$ samples per round (where $m$ is calculated based on the value of $s$ available to the algorithm). Furthermore, we allow GDSP to use $m = 57$ samples per round, which corresponds to $s = 15$, across all comparisons. We plot the excess cost that each algorithm accumulates over $T = 100$ rounds compared to exact gradient descent. We find that although the performance of CONGO-Z and CONGO-E is poor when the given $s$ is less than half the true $s$, their performance quickly approaches that of the baseline and does not deviate significantly as the given value of $s$ continues to increase. Thus we see that under the appropriate sampling strategy, the approach is robust to an inaccurate estimate of the sparsity. For CONGO-B,

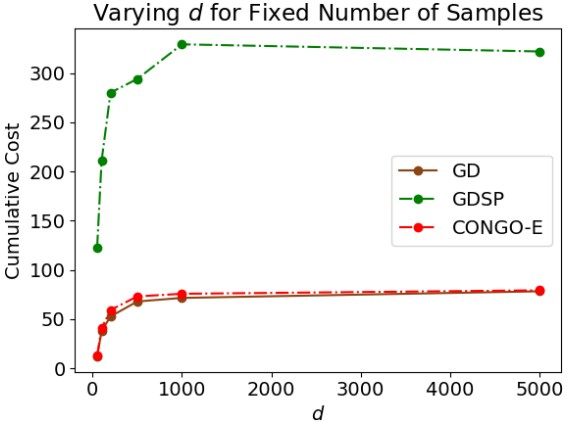

Figure 10: Cumulative costs incurred over 100 rounds for different values of $d$ with $s$ fixed at 5 and $m$ fixed at 24. Note how the cumulative cost of CONGO-E closely follows the cumulative cost of exact gradient descent, which represents the best we can hope to do.

the performance degrades more significantly as the given value of $s$ increases beyond the true value because a larger value of $m$ actually introduces greater noise into the gradient estimates which the greater averaging is not able to compensate for.

While there may not be much of a cost in performance for overestimating the value of $s$, there is still a cost in algorithm runtime due to taking more samples per round than is really needed. However, there is a potential for one to adaptively choose the number of measurements such that a desired level of accuracy in the gradient estimate is maintained without using many more samples than necessary. Cai et al. (2022) describes a method for adapting to the sparsity level designed for the stochastic optimization setting where the gradient in a previous round can provide useful information about the current gradient. This is not true in the online convex optimization setting, but a similar adaptive strategy could still be applied by starting with an optimistic estimate of $s$ and then adding on measurements if the error is believed to be large. Furthermore, the adaptation method relies on a proxy for the true gradient error, specifically $\frac{||\mathbf{A}^t\bar{\nabla} f_t(\boldsymbol{x}_t) - \boldsymbol{y}||_2}{||\boldsymbol{y}||_2}$, which will not necessarily be minimized by the true gradient since $\boldsymbol{y} = \mathbf{A}^t \nabla f_t(\boldsymbol{x}_t) + \frac{\delta}{2}\boldsymbol{x}_t^T \nabla^2 f_t \boldsymbol{x}_t$. However, by making $\delta$ suitably small we can mitigate that issue. This suggests that the adaptive method would remain useful in the OCO setting and reduce the difficulty of choosing the $s$ parameter for both CONGO-Z and CONGO-E (the sampling method used by CONGO-B is not compatible with this method). We leave the full analysis of this method in the OCO setting for future work.

### F.3  IMPACT OF LARGE PROBLEM DIMENSION UNDER A CONSTRAINED NUMBER OF SAMPLES

Although compressive sensing allows us to reduce the sample complexity for gradient estimation to merely a logarithmic dependence on $d$, for extremely high-dimensional problems it may remain problematic to continuously scale up the number of samples as $d$ increases. Here, we test whether $d$ can be increased without increasing $m$ for CONGO-E without seriously degrading the performance. We fix $m = 24$ which is the standard value we have used for $d = 50$ and $s = 5$, but let $d$ increase up to 5000. If we were to scale $m$ with $d$, this would require $m$ to increase up to 70. The results in Fig. 10 suggest that little is lost in terms of the empirical performance of CONGO-E relative to exact gradient descent when $d$ increases. Note however that this depends on $s$ remaining constant. We also show GDSP with the same number of samples for reference.

## G  DETAILS OF EXPERIMENTAL SETUP

### G.1  ALGORITHM IMPLEMENTATION DETAILS

In this section, we provide implementation details for both the CONGO-style algorithms and the baselines used in our experiments. Note that for the baselines, we distinguish between the instances

used for stochastic gradient descent on a latency function (Section 7) from those used for gradient descent on the quadratic functions (Section 6) since in the SGD case we opt to normalize gradients before taking the gradient descent step to improve stability. In the case of the CONGO-style algorithms, we use the same name for both to avoid confusion although this difference is present in their implementations as well. Finally, note that there are additional hyperparameters for each of the algorithms used on the real system, which are defined in Table 6. The code used to implement the algorithms can be found in the supplementary materials (or, in the case of CONGO-Z and CoSaMP, in the Github repository at https://github.com/caesarcai/ZORO).

**Gradient Descent:** Since we have a closed form for the true gradient of the quadratic functions considered in Section 6, we implement a standard projected gradient descent algorithm as an optimal baseline.

**Naive Stochastic Gradient Descent:** This zeroth-order algorithm applies a perturbation to each dimension of $x_t$ one at a time and uses the finite difference approximation (as in equation 9) to estimate the gradient on a per-dimension basis. This means that the number of samples for this algorithm is always $d + 1$.

**Gradient Descent/Stochastic Gradient Descent with Simultaneous Perturbations:** This is an implementation of Spall's SPSA (see Appendix A for details). Since this algorithm is able to estimate the full gradient after just two function evaluations, we average over several estimates so that the total number of samples matches that of the CONGO-style algorithms for a fair comparison.

**RL Agent Using Proximal Policy Optimization:** Proximal Policy Optimization (PPO) is a trust region based policy gradient algorithm that maximizes the following objective function

$$L\left(s, a, \theta_k, \theta\right) = \min \left( \frac{\pi_\theta(a \mid s)}{\pi_{\theta_k}(a \mid s)} A^{\pi_{\theta_k}(s,a),} \mathrm{clip} \left( \frac{\pi_\theta(a \mid s)}{\pi_{\theta_k}(a \mid s)}, 1 - \epsilon, 1 + \epsilon \right) A^{\pi_{\theta_k}}(s, a) \right) \quad (15)$$

This objective function ensures that the policy updates are not too drastic. Specifically, the gradient of the objective goes to zero if the ratio of the new policy probability to the old policy probability for any given state-action pair deviates more than $\epsilon$ away from 1. The PPO agent is trained on the DeathStarBenchmark Social Media environment (we do not use it in the Jackson network simulations) for 30 iterations to learn the impact of different CPU allocations on system performance. It is then tested for an additional 30 iterations to evaluate its effectiveness, which is what we compare to the other algorithms. PPO was chosen as a baseline to demonstrate the performance of an algorithm with prior system knowledge, unlike gradient descent approaches.

**CONGO-B:** In the numerical experiments for Section 6, we set $k_t = 3m$ of $k_t = 6m$ depending on the value of $d$ to ensure sufficient averaging. In the experiments for Section 7, we fix $k_t = m$ such that CONGO-B uses the same number of samples as CONGO-Z/E and GDSP/SGDSP. To implement the basis pursuit recovery method, we use Chambolle and Pock's primal-dual algorithm (see (Foucart and Rauhut, 2013), Section 15.2) for the Numerical and Jackson network experiments and the Sequential Least Squares Quadratic Programming (SLSQP) solver from the scipy optimize Python package for tests on the DeathStarBenchmark.

**CONGO-Z:** The code used to implement CONGO-Z comes directly from the code base for the paper which introduced ZORO (Cai et al., 2022). The only file that we needed to modify was optimizers.py, but the files Cosamp.py and base.py are also required.

**CONGO-E:** We provided our own implementation of CONGO-E, although it uses the same code for the CoSaMP algorithm as CONGO-Z does. The implementation is based on the pseudocode for Algorithm 1.

## G.2 NUMERICAL EXPERIMENTS FOR ONLINE OPTIMIZATION OF QUADRATIC FUNCTIONS

For all of the numerical simulation experiments, we used $T = 100$. Where applicable, the algorithms used the same hyperparameters. The value of $s$ provided to the algorithms was the exact value of $s$. These hyperparameters are listed in Table 1.

Table 1: Hyperparameters for Numerical Experiments

| Name | Description | Value |
|---|---|---|
| lr | learning rate | 0.1 |
| $\delta$ | smoothing parameter | $\begin{cases} 1e^{-5}, & \sigma = 0 \\ 0.05, & \sigma = 0.001 \end{cases}$ |
| $m$ | # rows in $\mathbf{A}^t$ | $\lceil 2s \log(\frac{d}{s}) \rceil$ |
| Recovery tolerance | Used for basis pursuit and CoSaMP | 0.005 |
| Recovery max iterations | Used for basis pursuit and CoSaMP | 50 |

Since the CONGO-style algorithms require bounds on $L_F \geq ||\nabla f_t||_2$ and $L \geq ||\nabla^2 f_t||_2$, we compute then for a given quadratic function as follows:

$$L_f = R\sqrt{2 \log(s)} + 2\sqrt{s}$$

$$L = \sqrt{2 \log(s)}$$

These are bounds in expectation; The value of $L_f$ comes from bounding $||\boldsymbol{D}\boldsymbol{x}||_2 + ||\boldsymbol{b}||_2$, where recall that $R$ is the radius of the constraint set (in this case centered on 0) and hence the maximum value that any entry of $\boldsymbol{x}$ can take. The value of $L$ comes from bounding the maximum of $s$ standard normal random variables in expectation since the eigenvalues of a diagonal matrix correspond to the diagonal entries.

For performing computations, the main Python package we use is Numpy (a full list of the packages required is available with the code). The seeds we used to obtain our results are 0-49. When creating our plots, we plot the average over the trajectories from the different seeds and also plot a shaded area representing $\pm$ one standard deviation away from the mean. The computation of the mean and standard deviation is done using functions provided by Numpy. The pyproximal package is also required by the code used for CONGO-Z.

### G.2.1 SPECIFICATIONS OF MACHINE USED

The numerical simulations were run on a machine with an Intel core i7 processor and an NVIDIA GeForce RTX 3050 Ti Laptop GPU. We found that each overall experiment with $d = 50$ did not take longer than two minutes to run.

### G.3 SIMULATION EXPERIMENTS FOR ONLINE OPTIMIZATION OF JACKSON NETWORKS

The following information concerns Section 7.1. We first provide a description of the layout of the Jackson networks and briefly explain how they are implemented, then describe the simulated workloads used, and finally provide the hyperparameter settings for the algorithm. Any additional information needed to setup and reproduce the experiments is provided with the code.

### G.3.1 JACKSON NETWORK LAYOUT

To simulate Jackson networks, we use the Python package queueing-tool (Jordon, 2023). Our simulation experiments use two different layouts, one which we call the complex environment and one which we call the large-scale environment. The first contains 15 queues and the second contains 50. For each layout we define a set of jobs, each of which is defined by an ordered sequence of queues which that job must be processed by. The length of this sequence varies from 1 to 7. All jobs share a single entry point which we label as "A". While the queueing-tool package comes with all the functionality needed to simulate a generic Jackson network, we found the need to implement a subclass of the Agent class to implement jobs with paths randomly chosen from a fixed set. However, this did not involve modifying the source code of the package.

A visual representation of the complex environment is given in 11. This environment has eight possible job types, named Job1 through Job8. The sequence of microservices that each type must visit is specified in the .yaml configuration files corresponding to the complex environment. The large-scale environment has a simpler structure; there are 10 job types, and after being processed by

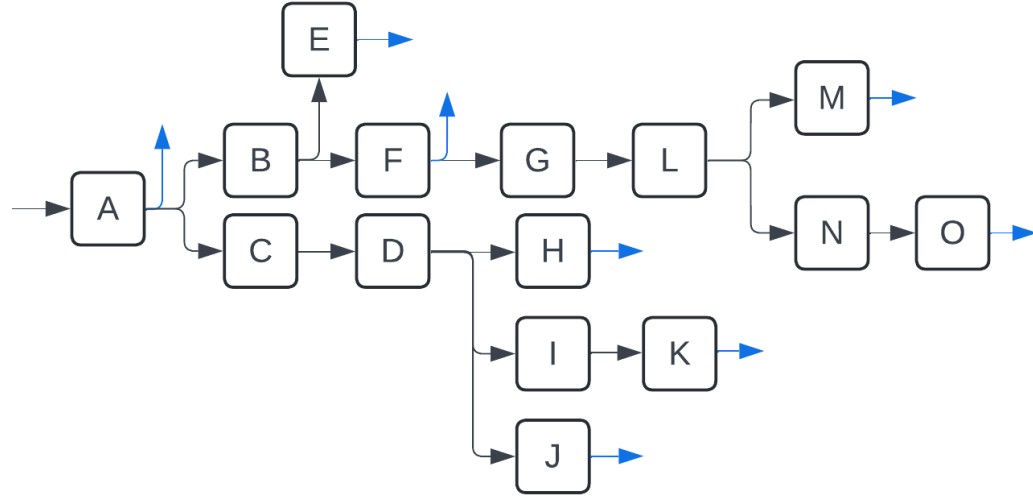

Figure 11: Visual representation of the connections between microservices for the 15-microservice environment. The labeled boxes correspond to microservices, the black arrows correspond to the entry point and the connections between microservices, and the blue arrows correspond to exit points. Each of the 8 possible jobs uses a different exit point.

the entry-point microservice each job moves through a unique sequence of four or five microservices before exiting the system. Hence, in the large-scale environment the job paths only overlap at the entry point.

In the queueing-tool framework, each microservice is represented as a pair of nodes with a queue edge inbetween. When a job leaves a microservice's queue, it instantaneously moves from the exit node of that microservice to the entry node of the next and is enqueued on the next microservice's queue. Using the functionality provided by the queueing-tool package, we generate arrivals according to a Poisson process. Each time a job arrival occurs, the new job has a job type assigned to it according to the workload distribution. The service times, and hence the measured latencies, are based on the Jackson network's simulation time. We advance the simulation time only when the algorithm incurs its cost for the round or takes one of its samples (thus nothing changes in the simulation while the algorithm is performing computations). When the average latency is measured, we first advance the simulation for 30 seconds to bring it to a steady state and then advance it for another 10 seconds while measuring the end-to-end latencies experienced by all jobs which leave the system during that period. At the end of a round, we use the clear() function provided by queueing-tool to reset the Jackson network to its initial state. We found this necessary because otherwise the Python object simulating the network would accumulate too much logged data which would make obtaining latency for newly processed jobs prohibitively slow. Running the simulation so that it reaches steady state before taking any measurements compensates for this, since the network is allowed to reach approximately the same state that it would have had under the new allocation without the reset.

### G.3.2 PROTECTING AGAINST INSTABILITY

To ensure that any algorithm has valid latency measurements to use when forming a gradient estimate, we monitor whether or not any jobs leave the system during a measurement period and if none do, we assume that the stability of the Jackson network has been violated. Regardless of which algorithm is in use, when this occurs we increase the allocation to every microservice by a small amount called the correction factor and proceed to the next round. This ensures that the algorithm is able to eventually recover from the instability even if it cannot form a gradient estimate. The choice of correction factor depends on the scenario; for the complex environment with fixed workload it is 1, for the complex environment with variable job type or arrival rate it is 0.5, and for all large-scale environment scenarios it is 0.1.

Besides the workload, there are a few other parameters which determine the environment that the algorithms operate in. These are the initial allocations for the microservices, the range of allowable allocations, and the resource weight (referred to as $w$ in the paper). For all experiments, the resource weight is set to 1 and the range of allowable allocations is the set [1, 60]. The initial allocations vary based on the experiment since in cases where the workload will change, we want the algorithms to have a chance to converge before the change becomes significant, forcing them to adjust. For simplicity, we keep this value uniform for all but the entry microservice (which requires a higher initial allocation to ensure stability in some cases).

Table 2: Initial Allocation by Configuration

| Configuration | For Entry Microservice | For Others |
|---|---|---|
| Complex, Fixed Workload | 10 | 10 |
| Complex, Variable Arrival Rate | 10 | 10 |
| Complex, Variable Job Type | 10 | 7 |
| Large-Scale, Fixed Workload | 7 | 7 |
| Large-Scale, Variable Arrival Rate | 7 | 7 |
| Large-Scale, Variable Job Type | 10 | 4 |

### G.3.3 DESCRIPTION OF WORKLOADS

The three workload types tested mimic the different workload types used in our experiments on the DeathStarBench Social Network Application. The first type is the fixed workload where both the average arrival rate and the job distribution remain constant for all rounds. The second type is variable arrival rate where the average arrival rate changes periodically while the job distribution remains constant. This forces the algorithm to adapt its allocation over time without directly changing the set of queues which process jobs. The third type is variable job type where the job distribution incrementally switches between two fixed distributions over a certain number of rounds; outside of those rounds, the job distribution is constant. This forces the algorithm to adapt to a change in which microservices require resources and involves a change in the sparsity during the transition period. In the following tables, we specify the parameters defining the workload used in each plot of 3.

Table 3: Parameters for Fixed Workloads

| Environment | Arrival Rate | Job Distribution |
|---|---|---|
| Complex | 5/s | Jobs 1,3,4,6,7,8: 2% Jobs 2,5: 44% |
| Large-scale | 5/s | Job6: 100% |

### G.3.4 HYPERPARAMETERS

**Learning Rate Schedules** Since our goal is to show that CONGO-E produces better gradient estimates than other algorithms under sparse conditions, we ensure that all algorithms use the same learning rate schedule so that no algorithm has a clear advantage in that respect. For both environment types we use the same initial learning rate of 1 for fixed workload and variable arrival rate scenarios, while for variable job type scenarios the initial learning rate is 0.7. For the the complex environment we decay the learning rate on every 25 rounds (with a factor of 0.7 for the fixed workload and variable arrival rate scenarios, and a factor of 0.5 for the variable job type scenario).

**Other Hyperparameters** For all algorithms we set $\delta = 0.5$ since making it too much smaller would cause noise to overwhelm the measurements; this applies equally for both the baselines and the CONGO-style algorithms since they all depend on measuring the difference in the function's value at two points, and if the noise in the function evaluations is larger than that difference then gradient estimation becomes ineffective. For the SGDSP and CONGO-B algorithms which only require two function evaluations to get a gradient estimate but average over many to reduce noise, we set the sample size for the average to the value of $m$ used by the CONGO-stype algorithms since this ensures

Table 4: Parameters for Variable Arrival Rate Workloads

| Environment | Arrival Rate | Job Distribution |
|---|---|---|
| Complex | Rounds 1-25: 5/s
Rounds 26-50: 5.5/s
Rounds 51-75: 6/s
Rounds 76-100: 5.5/s | Job6: 100% |
| Large-scale | Rounds 1-10: 4.5/s
Rounds 11-20: 4.75/s
Rounds 21-30: 5/s
Rounds 31-40: 5.25/s
Rounds 41-50: 5.5/s
Rounds 51-60: 5.25/s
Rounds 61-70: 5/s
Rounds 71-80: 4.75/s
Rounds 81-90: 4.5/s
Rounds 91-100: 4.75/s | Job2: 100% |

Table 5: Parameters for Variable Job Type Workloads

| Environment | Arrival Rate | Initial Distribution | Final Distribution | Transition Rounds |
|---|---|---|---|---|
| Complex | 5/s | Job 2: 50%
Job 5: 50% | Job 2: 10%
Job 3: 30%
Job 6: 60% | 40-90 |
| Large-scale | 4/s | Job 1: 50%
Job 3: 50% | Job1: 30%
Job3: 30%
Job6: 20%
Job8: 20% | 40-90 |

that they use the same number of samples as CONGO-Z and CONGO-E on each round. For the complex environment we use $s = 6$ and $m = 8$, while for the large-scale environment we use $s = 10$ and $m = 17$. For the primal-dual algorithm which implements basis pursuit as well as the CoSaMP algorithm, we set the tolerance to 0.005 and the maximum number of iterations to 50.

### G.3.5 SPECIFICATIONS OF MACHINE USED

Note that these simulations did not require the use of GPUs.

- **Operating System:** Ubuntu 22.04.3 LTS (Jammy)

- **Kernel Version:** 6.5.0-28-generic

- **CPU:** AMD Ryzen Threadripper 3960X 24-Core Processor

  - **Architecture:** x86_64
  - **Cores:** 12
  - **Threads:** 24
  - **Max Frequency:** 3.8 GHz

- **Memory:**

  - **Total Memory:** 96 GiB
  - **Swap:** 31 GiB

### G.4 DEATHSTARBENCH SOCIAL NETWORK APPLICATION

#### OVERVIEW

The DeathStarBench suite includes a social network application designed to evaluate microservices' performance. This application simulates real-world social network activities, providing a robust environment for benchmarking.

#### TESTING METHODOLOGY

The `wrk2` tool is utilized to generate workloads for the microservice application. This application consists of multiple containers, each hosting a different microservice. Integrated within these containers are Jaeger clients, which continuously record service-level metrics, specifically latency, in the background. These metrics are periodically sent to the Jaeger Collector.

The collected metrics serve as key performance indicators (KPIs) to assess the system's performance. The KPIs are then used to calculate the gradient, which informs the adjustments needed for CPU allocations. These adjustments are applied to the system's containers through Docker commands.

The specific methods for calculating the gradient and implementing CPU allocation modifications are algorithm-dependent. This adaptive approach ensures optimal resource allocation, thereby enhancing the overall efficiency and performance of the microservice application. 12 illustrates how algorithms interact with the social network application in our experiments.

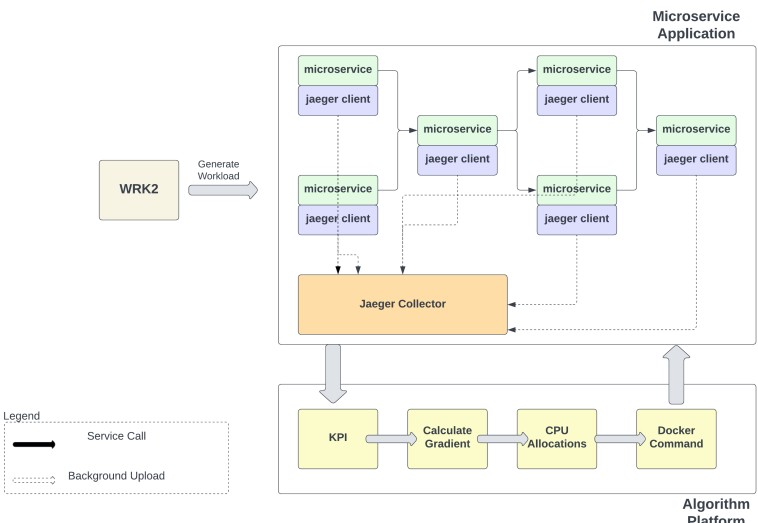

Figure 12: A visual representation of the application of algorithms within a microservice architecture, demonstrating how CPU allocations are adjusted to achieve optimal performance thresholds.

**Workloads:** We used the WRK2 tool to generate workloads for our experiments. We run the workload for a duration of 2 hours, with 4 client connections. We executed the benchmark against the following workloads:

- **Fixed Workload with fixed job types**:
  - Request per second = Fixed 2000 req/sec
  - Job type = compose-post
- **Fixed Workload with variable job types**:
  - Request per second = Fixed 2000 req/sec
  - Job type = composition of compose-post, read-hometimeline, read-usertimeline requests
- **Variable Arrival Rate with fixed job types**:
  - Request per second = follows this pattern [2000, 1900, 1500, 1600, 1800]
  - Job type = compose-post

### G.4.1 Hyperparameters

Each algorithm was tested with a specific set of hyperparameters to assess their performance under different conditions. Table 6 presents the glossary of the hyperparameters used for the various algorithms and Tables 7, 8, 9, 10,11, 11 provide the exact hyperparameters used for NSGD, SGDSP, PPO, CONGO-B, CONGO-Z and CONGO-E respectively.

Table 6: Glossary of Hyperparameters for Deathstar Bench Trials

| Hyperparameter | Definition |
| --- | --- |
| cpu period | Parameter for Docker to change CPU allocation (100000 for all) |
| max iterations | Maximum number of iterations for the algorithm (20 for all) |
| CPU_COST_FACTOR | Cost factor for CPU usage |
| latency_weight | Weight assigned to latency in the cost function |
| $\delta$ | Small value for CPU allocation adjustments |
| initial learning rate | Starting rate for updating CPU allocation |
| learning rate decay factor | Multiplier to reduce learning rate each iteration |
| update step limit | Max step size for CPU allocation updates |
| current cpu bounds | Range for allowable CPU usage |
| significance threshold for change | Min threshold for significant CPU allocation changes |
| learning rate decay schedule | Formula to decrease learning rate over time |
| number of calls (BO) | Iterations for Bayesian Optimization |
| acquisition function (BO) | Function to select next hyperparameters in BO |
| latency threshold | Maximum allowable latency for operations |
| momentum | Parameter to accelerate gradient vectors |
| exploration factor | Balance between exploring and exploiting strategies |
| n_steps | Number of steps per environment update in PPO |
| $\gamma_{PPO}$ | Discount factor in PPO |
| learning_rate | Learning rate in PPO |
| GAE Lambda (gae_lambda) | Controls bias-variance trade-off in advantage estimation (PPO) |
| Entropy Coefficient (ent_coef) | Encourages exploration by penalizing deterministic policies (PPO) |
| Value Function Coefficient (vf_coef) | Weight for the value function's error during training (PPO) |
| Max Gradient Norm (max_grad_norm) | Threshold for gradient clipping to prevent explosion (PPO) |
| $\gamma$ | $l1$-minimization error constraint (CONGO) |
| scale (sigmoid adjustment) | Scaling factor for sigmoid function to adjust CPU allocations (CONGO) |

Table 7: NSGD Hyperparameters for Deathstar Bench Trial

| Hyperparameter | Value |
| --- | --- |
| $\delta$ (gradient estimation) | 0.05 |
| initial learning rate | 0.01 |
| update step limit | 0.1 |
| current cpu bounds | 0.1 to 1.0 |
| significance threshold for change | 0.00009 |
| learning rate decay schedule | 1.0 / (1.0 + decay_factor * iteration) |
| CPU_COST_FACTOR | 10.0 |
| latency_weight | 1.0 |

Table 8: SGDSP Hyperparameters for Deathstar Bench Trial

| Hyperparameter | Value |
| --- | --- |
| $\delta$ | 0.02 |
| learning_rate | (1.0 / (1.0 + steps)) |
| CPU_COST_FACTOR | 10.0 |
| latency_weight | 1.0 |

Table 9: Proximal Policy Optimization (PPO) for Deathstar Bench Trial

| Hyperparameter | Value |
| --- | --- |
| $\delta$ | 0.05 |
| max iterations | 60 (30 - training, 30 -test) |
| n_steps | 2048 |
| $\gamma_{PPO}$ | 0.99 |
| learning_rate | 0.015 |
| gae_lambda | 0.95 |
| ent_coef | 0.01 |
| vf_coef | 0.5 |
| max_grad_norm | 0.5 |
| CPU_COST_FACTOR | 10.0 |
| latency_weight | 1.0 |

### G.4.2 SPECIFICATIONS OF MACHINE USED

- **Operating System:** Ubuntu 22.04.3 LTS (Jammy)
- **Kernel Version:** 6.5.0-28-generic
- **CPU:** Intel(R) Core(TM) i9-9940X CPU @ 3.30GHz
    - **Architecture:** x86_64
    - **Cores:** 14
    - **Threads:** 28
    - **Max Frequency:** 4.5 GHz
- **Memory:**
    - **Total Memory:** 62 GiB
    - **Swap:** 31 GiB
- **GPU:**
    - 2 x NVIDIA GeForce RTX 2080 Ti

Table 10: CONGO - B Hyperparameters for Deathstar Bench Trial

| Hyperparameter | Value |
|---|---|
| number of measurements ($m$) | 12 |
| $\delta$ | 0.2 / max perturbation value |
| $\gamma$ | 0.2 / (1 + iteration * 0.05) |
| learning_rate | 0.6 / (1 + iteration * 0.5) |
| scale (sigmoid adjustment) | 5 |
| CPU_COST_FACTOR | 10.0 |
| latency_weight | 1.0 |

Table 11: CONGO - Z Hyperparameters for Deathstar Bench Trial

| Hyperparameter | Value |
|---|---|
| $\delta$ (perturbation size) | 0.1 |
| learning rate | 0.15 |
| sparsity ($s$) | 6 |
| number of measurements ($m$) | 10 |
| latency weight | 1.0 |
| CPU allocation weight | 10.0 |

Table 12: CONGO - E Hyperparameters for Deathstar Bench Trial

| Hyperparameter | Value |
|---|---|
| $\delta$ (perturbation size) | 0.1 |
| learning rate | starts at 0.1, decreases as $0.1/(1 + \text{iteration} \times 0.1)$ |
| sparsity ($s$) | 6 |
| number of measurements ($m$) | 10 |
| initial CPU allocation | 0.35 |
| latency weight | 1.0 |
| CPU allocation weight | 10.0 |

