# OpenReview forum: "CONGO: Compressive Online Gradient Optimization"
_ICLR.cc/2025/Conference — ICLR 2025 Poster_

### Official Review · Reviewer_zPCR · 2024-10-16

**Soundness:** 3
**Presentation:** 2
**Contribution:** 3
**Rating:** 6
**Confidence:** 2

**Summary:**

The paper introduced the compreesive online gradient optimization framework (CONGO) for solving sparse online convex optimization problems based on motivating examples in real world application. Three variants using simultaneous perturbation and compressive sensing are proposed based on different inspiration. The proposed algorithms were validated both theoretically and experimentally.

**Strengths:**

1. The idea of using compressing sensing to estimate gradients more efficiently is novel. From the theory provided, the total cost of function value evaluation is reduced which makes it suitable for problems of large dimension.

2. Based on the specific setting and the measurements used, three different variants are proposed, which makes the framework flexible.

3. The paper presents theory on the regret bound and validate the theory with experiments.

**Weaknesses:**

While the paper is generally interesting and the idea is novel, the reviewer wants to point out that the paper contains *formatting errors*, specifically, the heading "Under review as a conference paper at ICLR 2025" is missing at each page.

1. It seems to the reviewer that the proposed method needs careful tuning of the parameters, such as the step size and the sparsity, which may require additional information which makes the proposed algorithm less practical.

2. CONGO-B seems to be less efficient and more unstable compare to the other variants, this is not explained in theory.

3. The proposed assumptions are a little restrictive in reality.

**Questions:**

1. How sensitive is the proposed algorithm to sparsity? What would happen if the sparsity measure $s$ is large, will the performance still be comparable to other existing methods?

2. Could adaptive strategies be used for the sparsity?

---

> ### Author Response · Authors · 2024-11-20
>
> We thank the reviewer for their careful review and consideration. We hope that our responses below provide clarification and are ready to discuss these points further as needed.
>
> [Weakness 0] [*Formatting issue*]
> We apologize for this oversight on our part. The formatting on the submission has been corrected.
>
> [Weakness 1/Question 1] [*Discussion on parameter choices*]
> Numerical simulations have proved CONGO-Z and CONGO-E to be reasonably robust to the choice of $s$ and $m$. To augment our original results in Appendix F discussing the robustness of CONGO-E to the value of $m$, we have added empirical results on the impact of assuming a value for $s$ that is inaccurate (see Section F.2). Some parameters like $\delta$ and the learning rate are necessarily problem-dependent, with the optimal values depending on things like the presence of noise in function evaluations and the smoothness of the objective function.
>
> [Weakness 2] [*Efficiency of CONGO-B versus other CONGO variants*]
> The CONGO framework can generate a variety of algorithms, and not all of them will necessarily be equally effective. Our intention was to display multiple design choices that fit the CONGO framework while proving that there exist some CONGO algorithms that are highly efficient. Thus, we do not view the particular weaknesses of CONGO-B as weaknesses of the CONGO framework as a whole. We have added a remark to Section 4 (page 6) which clarifies this.
>
> [Weakness 3] [*Restrictiveness of assumptions*]
> As we acknowledged in our response to Reviewer qS2T, the assumption of exact gradient sparsity seems to be necessary for an algorithm explicitly using sparse gradient estimates to achieve sublinear regret in the adversarial setting. There is a limit to how close a sparse gradient estimate can get to a non-sparse true gradient, and if the average error in the gradient estimate cannot be made to decrease continually as $T$ increases then linear regret is inevitable. In terms of empirical performance, however, we have added new experiments in section F.2 which show the impact of weakening the exact gradient sparsity assumption.
>
> Furthermore, our assumption that the algorithm can make several measurements on each round seems reasonable for the applications we consider in our paper, and has been used in previous work on zeroth-order online convex optimization. The slow evolution of the objective function for autoscaling microservices has been observed in [1], where it was found that in 55\% of the microservice application deployments studied, the load did not change by more than 10\% over a five minute interval. In our autoscaling experiment in Section 7.2, that would give more than enough time to do a full gradient estimate. The other assumptions we make on the objective function in Section 2 are standard in the constrained OCO literature.
>
> [1] Luo, S.; Xu, H.; Lu, C.; Ye, K.; Xu, G.; Zhang, L.; Ding, Y.; He, J.; Xu, C. Characterizing microservice dependency and performance: Alibaba trace analysis. In Proceedings of the ACM Symposium on Cloud Computing, Seattle, WA, USA, 1–4 November 2021
>
> [Question 2] [*Adaptive strategies for sparsity parameter*]
> An adaptive strategy motivated by the one proposed by Cai et. al. [2] could be applied by starting with an optimistic estimate of $s$ and then adding on measurements if the error is believed to be large. We believe this could be integrated into CONGO-Z or CONGO-E without major computational overhead to reduce the difficulty of choosing the $s$ parameter. We have added a discussion on this matter to pages 24-25.
>
> [2] HanQin Cai, Daniel McKenzie, Wotao Yin, and Zhenliang Zhang. Zeroth-order regularized optimization (zoro): Approximately sparse gradients and adaptive sampling. SIAM Journal on Optimization, 32(2):687–714, April 2022.

---

> > ### Comment · Reviewer_zPCR · 2024-11-26
> >
> > Thanks for the detailed clarifications!

---

### Official Review · Reviewer_qS2T · 2024-10-30

**Soundness:** 3
**Presentation:** 3
**Contribution:** 3
**Rating:** 8
**Confidence:** 4

**Summary:**

This paper introduces a framework designed for zeroth-order online convex optimization (OCO) in environments where gradients are often sparse. The core idea is to combine compressive sensing techniques with online optimization to take advantage of this sparsity, which allows for efficient high-dimensional optimization with fewer samples.

The authors propose three variations—CONGO-B, CONGO-Z, and CONGO-E—that utilize different compressive sensing approaches and gradient estimation methods. They back up their approach with theoretical guarantees, showing sublinear regret bounds, and validate the framework through experiments on both synthetic and real-world tasks (like microservice autoscaling).

**Strengths:**

1. The use of compressive sensing within an OCO framework is a fresh and well-motivated idea. By focusing on sparse gradients, the authors address both sample efficiency and dimensionality reduction, which are critical in high-dimensional settings.

2. The authors provide rigorous theoretical analysis, establishing regret bounds that demonstrate sublinear scaling with respect to the problem horizon, independent of the problem dimension.

3. The three algorithmic variants, CONGO-B, CONGO-Z, and CONGO-E, offer a nice balance of performance and complexity. For instance, CONGO-E uses Gaussian matrices and CoSaMP for enhanced performance, while CONGO-Z is more sample-efficient.

**Weaknesses:**

1. The theoretical analysis assumes exact gradient sparsity, which may not be realistic for all real-world problems.

2. While CONGO outperforms standard gradient descent with SPSA, it’s mostly compared against methods that don’t leverage sparsity. A more comprehensive comparison with advanced sparse optimization techniques or regularized gradient estimators would help here.

**Questions:**

1. The theoretical analysis assumes exact gradient sparsity, which may not hold in all practical scenarios. Could the authors discuss potential extensions of their theoretical analysis to approximately sparse gradients, or provide insights on how performance changes as the level of sparsity decreases? This could help clarify the framework's robustness and practical applicability.

2. While the regret bounds are dimension-independent, sample complexity grows logarithmically with dimension. In very high-dimensional settings, how does this sample complexity impact practical performance?

3. The framework is compared to standard SPSA-based methods, but how does it compare to other advanced sparse optimization or regularized gradient estimation techniques?

---

> ### Author Response · Authors · 2024-11-20
>
> We thank the reviewer for their extensive analysis and hope that our responses below provide clarification. We are ready to discuss these points further as needed.
>
> [Weakness 1/Question 1] [*The need for exact gradient sparsity*]
> The assumption of exact gradient sparsity is necessary in the adversarial setting to enable an algorithm which uses sparse gradient estimates to achieve sublinear online regret. Without that assumption, there would be a lower bound on the gradient error at each round which would lead to a linear increase in the regret so long as the value of the gradient along more than $s$ dimensions does not decay with $T$. The purpose for applying our algorithm in realistic scenarios like the microservice autoscaling problem is to show that empirically the assumption of exact gradient sparsity is not required for CONGO-style algorithms to outperform algorithms which do not exploit sparsity. To make this more clear, we have added the results from new numerical experiments in Section F.2 of the appendix which demonstrate the robustness of CONGO-Z and CONGO-E to a misspecification in the value of $s$ given to the algorithm and their robustness when the ratio $\frac{d}{s}$ is smaller than what we considered before.
>
> [Weakness 2/Question 3] [*Comparisons are mostly with respect to methods that do not leverage sparsity*]
> We note that, to the best of our knowledge, there is no prior work on OCO which is explicitly designed to exploit sparse gradients, which is why we have mostly compared different versions of CONGO to evaluate their relative merits. Part of our rationale for introducing multiple CONGO variants is to show that the CONGO variant which we believe to be most effective, CONGO-E, indeed uses the most efficient sparse gradient estimation scheme for OCO (rather than the one introduced in [1], for example, which is the inspiration for CONGO-B). Please also see our response to reviewer LC7u's Weakness 1, which addresses a similar concern.
>
> [1] Vivek S. Borkar, Vikranth R. Dwaracherla, and Neeraja Sahasrabudhe. Gradient estimation with simultaneous
> perturbation and compressive sensing. Journal of Machine Learning Research, 18(161):1–27, 2018
>
> [Question 2] [*Practical implications of logarithmic growth of sample complexity with dimension*]
> We have added numerical experiments for very high-dimensional cases (up to $d = 5000$) in Section F.3 of the appendix. These experiments test what happens if the dimension $d$ increases without increasing the number of samples per round to compensate, and we find that CONGO-E is able to maintain similar performance to gradient descent with full gradient information despite not using the theoretically correct number of samples. This suggests that the sample complexity actually required for good performance will not become prohibitively large in very high-dimensional settings as long as gradients are appropriately sparse. Note that it is not practical for us to consider such high-dimensional problems for the Jackson network and microservice autoscaling experiments due to limited computational resources.

---

### Official Review · Reviewer_LC7U · 2024-10-30

**Soundness:** 3
**Presentation:** 3
**Contribution:** 3
**Rating:** 6
**Confidence:** 3

**Summary:**

This paper proposed a framework for online zeroth-order optimization leveraging the techniques and insights from compressed sensing. The authors provide several schemes for efficiently sampling the objective functions values and estimate the gradients, alongside with theoretical convergence proofs revealing the fast convergence rates. The numerical results demonstrate this approach's superior performance over state-of-the-art baselines.

**Strengths:**

This is a well-written paper in general, the idea of introducing compressed sensing for estimating the gradients is very inspiring. The numerical performance of the proposed scheme is excellent. The presentation of the paper is very clear and easy to read.

**Weaknesses:**

The novelty of the proposed scheme may be potentially limited (rebuttal against this point is welcomed as the reviewer is not familiar with zeroth-order optimization literature). The reviewer has seen similar approach been proposed in Wang et al, "Stochastic zeroth-order optimization in high dimensions" AISTATS'18, where they utilized a very similar idea but used LASSO (L_1) instead of CoSAMP (L_0). The numerical study did not considered this AISTATS'18 paper as a baseline, although being cited in the reference.

**Questions:**

Could the authors clarify the fundamental difference between the proposed method and AISTATS'18 paper mentioned above? Please highlight the advantages of the proposed method.

In the numerical experiments, did you observe any phase transition on the number of measurements to the convergence speed of the proposed algorithms?

---

> ### Author Response · Authors · 2024-11-20
>
> We thank the reviewer for their praise and insightful comments, and we hope to hear their thoughts about our responses below.
>
> [Weakness 1/Question 1] [*Clarifying our contributions with respect to the results from AISTATS'18*]
> The gradient estimation procedure used in the AISTATS'18 paper does not exactly match any of the methods we consider, and its theoretical guarantees appear to be inferior to those which CONGO-Z and CONGO-E enjoy. We have added a discussion of this to Appendix A. More importantly, the fundamental differences between their problem setting and ours make it so that the algorithms proposed in that paper are not directly suitable for the goal of achieving sublinear OCO regret in a sample-efficient way. On one hand, the successive component selection algorithm (Algorithm 2) is not applicable to OCO at all. It relies on the assumption that the value of the objective function will never depend on most of the dimensions of the input. The mirror descent algorithm (Algorithm 3), on the other hand, assumes that each gradient is being taken using $\Theta(\sqrt{T})$ samples, which makes it much less reasonable to assume that the objective function remains fixed throughout the sampling process. Furthermore, it is not obvious that the probability of failure for their gradient estimation method can be made to decrease as $T$ increases, which we have shown is necessary to achieve a sublinear expected regret.
>
> [Question 2] [*Phase transition for convergence rate in terms of number of measurements*]
> In Appendix F (particularly Figure 6) we showed the effect of reducing $m$ (the number of measurements for CONGO-E) on the error in the gradient estimate. In that case, we had observed that setting $m = 24$ gave very good results, but it could be reduced to 21 without much of a loss in accuracy. However, once $m$ reached about 2/3 of the starting value, the error began to increase dramatically. This seems indicative of such a phase transition.

---

> > ### Comment · Reviewer_LC7U · 2024-11-25
> >
> > Many thanks for your clarification!

---

### Official Review · Reviewer_YSr4 · 2024-11-04

**Soundness:** 3
**Presentation:** 3
**Contribution:** 3
**Rating:** 8
**Confidence:** 3

**Summary:**

The paper studies zeroth-order online convex optimization, where the gradient of the objective function is assumed to be sparse. The proposed algorithms, CONGO, combine the (projected) gradient descent algorithm for online convex optimization, with a gradient estimation procedure using compressive sensing technique. The regret is proven to be O(\sqrt(T)) and does not depend on the dimension of the problem, and the per-iteration sampling complexity scales with the sparsity level of the gradient. Experiments confirm the effectiveness of the algorithms.

**Strengths:**

The paper is overall well written, and presents the setup, results, and proof clearly. The algorithms proposed appear efficient in terms of the regret and the sampling complexity.

**Weaknesses:**

-- One might argue that the results are not too surprising: the regret follows from the regret of online gradient descent, while the sampling complexity follows from the compressive sensing results.

-- In CONGO-B, line 827 – 829, gradient recovery requires solving an LP, which can be computationally inefficient, especially in high-dimensional setting. In addition, compressive sensing usually requires knowledge of the sparsity level before setting the number of samples. If such knowledge is lacking or inaccurate, compressive sensing might fail completely [1].

[1] Amelunxen, Dennis et al. “Living on the edge: phase transitions in convex programs with random data.” Information and Inference: A Journal of the IMA 3 (2013): 224-294.

**Questions:**

The regrets in Theorem 2 and 3 holds in expectation, but if understanding is correct, they also hold with high probability?

---

> ### Author Response · Authors · 2024-11-20
>
> We thank the reviewer for their feedback and invite them to continue to discuss these points based on our responses below.
>
> [Weakness 1] [*Main challenges and positioning of results*]
> The main theoretical challenge in applying compressive sensing to online convex optimization is ensuring that the error due to compressive sensing failures does not accumulate over $T$ (which would lead to a large regret). The main challenge in the practical implementation of CONGO within the framework from section 4 is finding the best choices for the measurement matrix and recovery algorithm to achieve good speed and performance. We believe that we found superior choices in CONGO-E, which does not exactly match any existing gradient estimation method (though it is inspired by multiple).
>
> [Weakness 2] [*CONGO-B gradient recovery requires solving an LP*]
> This is one of the reasons why we consider CoSaMP to be the better choice of recovery algorithm for other CONGO variants; however, to better show the range of possible options under the CONGO framework, we used basis pursuit for CONGO-B which aligns with the choice in Borkar et. al. for the stochastic optimization setting.
>
> [Weakness 3] [*Compressive sensing usually requires knowledge of the sparsity*]
> This is indeed true.  However, we have now included empirical results demonstrating that under the appropriate sampling strategy (used in CONGO-Z and CONGO-E), the approach is robust to an inaccurate estimate of the sparsity in the sense that it will still perform better than Spall's SPSA even if $s$ is much larger than the predicted value (see Section F.2 on pages 23-25). In fact, those results show that even when there is a 50\% error in the estimate of $s$, the performance will not be too much worse that gradient descent with full information.
>
> [Question 1] [*High probability equivalents of Theorems 2 and 3*]
> One could derive high-probability bounds in much the same way as the bounds in expectation for a version of the algorithms in which a single measurement matrix is drawn at the beginning and reused at every round. In such a case, the law of total expectation would no longer be applied because we would only be considering outcomes where the gradient estimation error is bounded as in Lemma 2. The probability with which the bound holds would be equal to epsilon in Lemma 2, which could be chosen freely as long as m is set appropriately. Thus, it would not be necessary for $m$ to depend on $T$. However, as stated in Appendix D, we opt to resample on every round for practical reasons (to ensure that no single trajectory experiences a very large number of failures), and so we did not include theorems for the high-probability bounds.

---

> > ### Comment · Reviewer_YSr4 · 2024-11-25
> >
> > Thank the authors for the clarifications and the additional experiments! I've raised my score.

---

### Author Response · Authors · 2024-11-20
**Message to All Reviewers**

We are glad that the reviewers found the paper well written/interesting with clear proofs (all reviewers), that the idea of compressive sensing for OCO is novel (zPCR), very inspiring (LC7U), and is a fresh and well-motivated idea (qS2T).  Reviewers have also remarked that the algorithms show a nice balance of performance and complexity (qS2T) and that the numerical performance is excellent (LC7U).

We have addressed each of the reviewer comments in individual responses.  We also have conducted additional experiments and have updated the paper.  The changes based on reviewer feedback mostly appear in the appendix and are indicated in blue text. Please do let us know if we can do anything further to improve the paper.

---

### Meta-Review · Area_Chair_AFEq · 2024-12-16

**Metareview:**

This paper focuses on zeroth-order online convex optimization under the assumption that the objective function's gradient is sparse. The proposed algorithms, CONGO, incorporate a compressive sensing-based gradient estimation procedure into the (projected) gradient descent framework for online convex optimization. By appropriately leveraging sparsity, the paper derives regret bounds that exhibit optimal dependence on the time horizon while maintaining favorable scaling with the problem dimension.

Most reviewers find the paper well-written, presenting numerous novel ideas and making a significant contribution to the literature on online optimization.

**Additional Comments On Reviewer Discussion:**

Although questions were raised regarding its relation to prior work, the role of hyper-parameters, and the sensitivity to the level of sparsity, the authors appear to have addressed many of these concerns.

---

### Decision · Program_Chairs · 2025-01-22

Accept (Poster)